# MAMBO-G: Magnitude-Aware Mitigation for Boosted Guidance

**Shangwen Zhu** [* 1]  **Qianyu Peng** [* 2]  **Zhilei Shu** [* 3]  **Yuting Hu** [1]  **Han Zhang** [1]  **Andy Zheng** [4]  **Xinyu Cui** [5]
**Jian Zhao** [6]  **Ruili Feng** [‡ 4]  **Fan Cheng** [† 1]

Github: https://github.com/xiaomomy/MAMBO-G
Project Homepage: https://matrixteam-ai.github.io/MatrixTeam-OmniVeritas/blog/mambo-g/

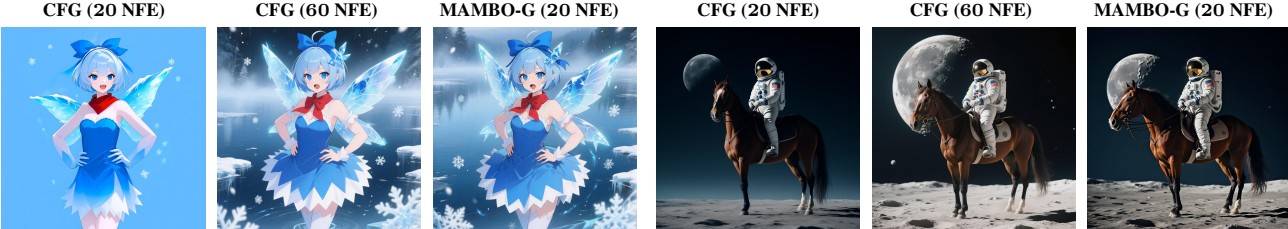

(a) Cirno from the Touhou Project (Seed 0 in Qwen-Image).          (b) Classic astronaut riding a horse (Seed 0 in Qwen-Image).

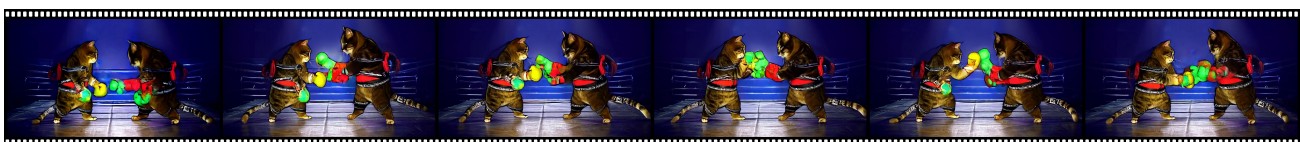

(c) CFG(30 NFE). Two cats fight on a spotlighted stage (Seed 0 in Wan2.2-5B, prompt from wan2.2 example).

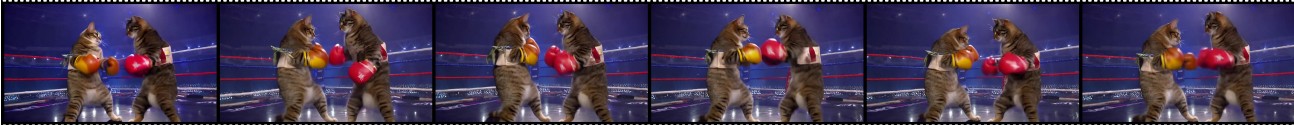

(d) **MAMBO-G** (30 NFE). Two cats fight on a spotlighted stage (Seed 0 in Wan2.2-5B).

*Figure 1.* **Superior efficiency of MAMBO-G :** Our method achieves comparable quality to 60-NFE (30-step) CFG image generation with only 20 NFE (10 steps), demonstrating a $3.0\times$ speedup over the standard CFG sampling (guidance scale = 4.0). The examples we demonstrate are **not cherry-picked**, and the seeds are also marked. Specific prompts can be found in Section A.4.

## Abstract

High-fidelity text-to-image and text-to-video generation typically relies on Classifier-Free Guidance (CFG), but achieving optimal results often demands computationally expensive sampling schedules. In this work, we propose **MAMBO-G**, a training-free acceleration framework that significantly reduces computational cost by dynamically optimizing guidance magnitudes. We observe that standard CFG schedules are inefficient, applying disproportionately large updates in early steps that hinder convergence speed. **MAMBO-G** mitigates this by modulating the guidance scale based on the update-to-prediction magnitude ratio, effectively stabilizing the trajectory and enabling rapid convergence. This efficiency is particularly vital for resource-intensive tasks like video generation. Our method serves as a universal plug-and-play accelerator, achieving up to $3\times$ speedup on Stable Diffusion v3.5 (SD3.5) and $4\times$ on Lumina. Most notably, **MAMBO-G** accelerates the 14B-parameter Wan2.1 video model by $2\times$ while pre-

---

[*]Equal contribution [‡]Project Leader [†]Corresponding author.
[1]Shanghai Jiao Tong University [2]The University of Hong Kong [3]University of Science and Technology of China [4]University of Waterloo [5]Chinese Academy of Sciences [6]Zhongguancun Academy. Correspondence to: Fan Cheng <chengfan85@gmail.com>.

*Proceedings of the $43^{rd}$ International Conference on Machine Learning*, Seoul, South Korea. PMLR 306, 2026. Copyright 2026 by the author(s).

serving visual fidelity, offering a practical solution for efficient large-scale video synthesis. Our implementation follows a mainstream open-source diffusion framework and is plug-and-play with existing pipelines.

# 1. Introduction

Generative models have made significant progress in creating images and videos from text (Song et al., 2021; Dhariwal & Nichol, 2021; Peebles & Xie, 2023; Ho et al., 2022; Esser et al., 2023). A key technique they use is classifier-free guidance (CFG) (Ho & Salimans, 2021), which adjusts the model's output to better match the text prompt. However, using strong guidance can sometimes reduce stability, potentially leading to issues like oversaturated colors or unnatural structures (Karczewski et al., a; Sadat et al., 2024). These problems are often more noticeable in modern, high-dimensional models, where using guidance scale without careful adjustment may strongly affect visual quality.

Recent studies have analyzed the stability and mechanisms of guidance strategies (Lin et al., 2024; Wang et al., 2025). As models scale to latent spaces with millions of dimensions, they encounter challenges associated with high-dimensional spaces. In such environments, the initial noise magnitude naturally scales with dimensionality (De Bortoli et al., 2022). Our analysis indicates that specifically at the initial timestep of generation, guidance update (difference between the conditional and unconditional model outputs) shares a similar direction across samples. In high-dimensional settings, forcing such a generic direction with a large guidance scale across diverse initial noises can destabilize the early generation trajectory, leading to severe overshooting and deviation from the realistic data distribution.

To address this, we propose **MAMBO-G**. This method automatically adjusts the guidance scale by comparing it to the model's inherent denoising activity. When guidance is very strong relative to the denoising process, **MAMBO-G** temporarily reduces the guidance scale, which helps keep the generation process stable in the early stages, while allowing a larger guidance scale later when the tones and structures of the image are clearer. The method is designed to be simple with almost no additional computational overhead, and to be compatible with various existing models and other CFG optimization strategies.

Our adaptive guidance schedule aims to accelerate conditional generation while maintaining sample quality, as suggested by metrics like ImageReward, CLIPScore, and vBench. Experiments indicate that **MAMBO-G** can achieve faster inference on models such as SD3.5 (Esser et al.), Lumina (Gao et al., 2024), and Wan2.1-14B (Team, 2025) compared to baselines. Specifically, results show up to $3\times$ acceleration on SD3.5, $4\times$ on Lumina, and $2\times$ on Wan2.1-14B, while achieving comparable or better performance than their slower baselines. The method also appears robust to hyperparameter settings and applicable across different model scales and domains. Moreover, **MAMBO-G** is orthogonal to other guidance optimization methods, such as Guidance Rescale (Lin et al., 2024) and Adaptive Projection Guidance (Sadat et al., 2024), enabling seamless integration for cumulative benefits.

In summary, our main contributions are as follows:

- We analyze the impact of early-step guidance in flow-based models, supported by theoretical motivation and empirical observations.

- We introduce a practical, magnitude-aware adaptive guidance schedule that aims to balance guidance scale across sampling steps.

- Through experiments on image and video generation, we demonstrate that our approach can facilitate speedups while maintaining the quality compared to standard CFG, with broad compatibility.

**Conflict of Interest Disclosure.** The authors declare no financial conflicts of interest. The models evaluated in this work, including SD3.5, Lumina, and Wan2.1-14B, are not developed by any organization that employs the authors. This research was conducted solely within academic institutions and received no funding from commercial entities with a stake in the reported results.

# 2. Related Work

## 2.1. Guidance Strategies in Diffusion Models

Various methods aim to improve guidance in diffusion models. **CFG++** (Chung et al.) treats guidance as a manifold-constrained inverse problem (Karczewski et al., 2026), while **CFG Schedulers** (Xi et al.) and **Apply Guidance in Interval** (Kynkäänniemi et al., 2024) optimize time-dependent strength; similarly, **Stage-Wise Dynamics** (Jin et al., 2025) investigates varying guidance requirements across different generation stages. **ReCFG** (Xia et al., 2025) and **TFG** (Ye et al., 2024) focus on correcting expectation shifts, a goal shared by **Rectified CFG++** (Saini et al., 2025) and **CFG-EC** (Yang et al., 2025) which propose mechanisms to rectify guidance errors and improve consistency. Recent approaches explore dynamic adaptation: $\mathbf{S^2}$**-Guidance** (Chen et al., 2025) uses stochastic block-dropping, **REG** (Gao et al., 2025) optimizes scaled joint distributions, **FBG** (Koulischer et al., 2025) uses feedback on conditional informativeness, and **Foresight Guidance** (Wang et al., 2025) frames CFG as a fixed-point iteration with short inner loops. Complementary

Resolution: 256        Resolution: 512        Resolution: 768        Resolution: 1024

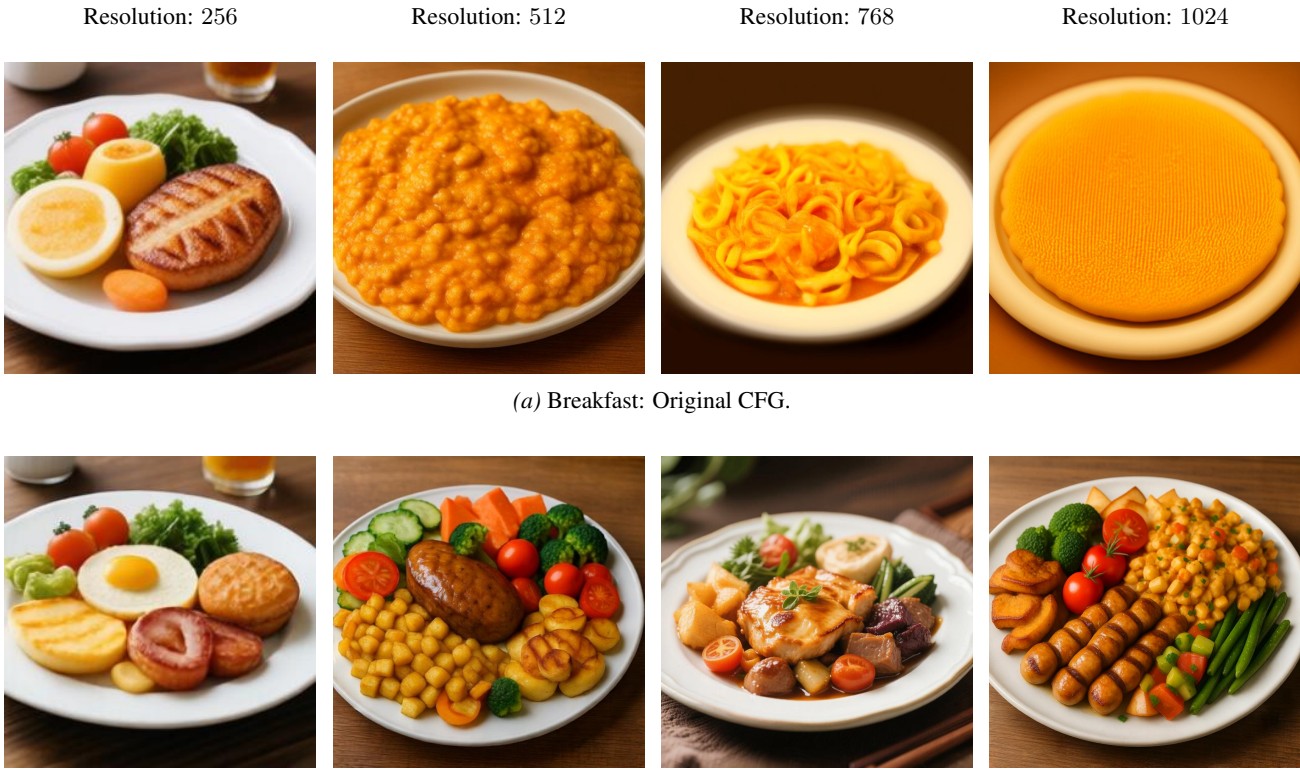

*(a)* Breakfast: Original CFG.

*(b)* Breakfast: **MAMBO-G** .

*Figure 2.* **Visual comparison across resolutions**: These are Qwen-Image (Wu et al., 2025) 10-step samples. From the results, it can be seen that with the original CFG, the higher the resolution, the more unstable the model's sampling results are. With **MAMBO-G** , our method stabilizes the sampling process by adjusting the guidance scale at the instance-level, showing significant improvements.

to these, **Prompt-Aware Guidance** (Zhang & Li, 2025) adapts strength based on prompt complexity, **Learning-to-Guide** (Galashov et al., 2025) employs meta-learning for optimal strategies, and **Saddle-Free Guidance** (Yeats et al., 2025) navigates the optimization landscape to avoid saddle points. **Density Guidance** (Karczewski et al., b) extends flow matching by incorporating explicit log-density control to steer generation trajectories. While it offers a rigorous theoretical unification of prior heuristics via Score Alignment, the method is computationally expensive. Specifically, estimating the divergence requires Jacobian-Vector Products (JVP), which doubles the inference cost and introduces estimation variance. In Flow Matching models (Esser et al.; Lipman et al., 2023; Fan et al., 2025a; Liu et al., 2023; Gao et al., 2024), **CFG-Zero\*** (Fan et al., 2025b) compensates for velocity errors.

### 2.2. Challenges of Zero-SNR Sampling.

Lin et al. (2024) observe that guidance becomes unstable when sampling from true zero-SNR, attributing this to excessive update magnitudes. While Rectified Flow models (Liu et al., 2023) resolve the training-inference mismatch of standard diffusion schedules by explicitly enforcing a pure Gaus-

sian boundary, they remain susceptible to the zero-SNR guidance instability identified by Lin et al. (2024). In this work, we analyze why early guidance update leads to instability. Based on this analysis, we propose an adaptive magnitude control mechanism to effectively stabilize the guidance trajectory.

### 2.3. Challenges in Large-Scale Generative Models

Early diffusion models, such as Stable Diffusion v2 (Rombach et al., 2022), operated on relatively compact latent spaces ($\approx 1.6 \times 10^4$ dimensions). In these settings, standard guidance strategies proved robust and forgiving to hyperparameter choices.

However, the transition to modern DiT-based architectures involves a massive increase in scale. For example, Flux (Labs, 2025) (generating 2K resolution) involves $\approx 10^6$ dimensions, and video models like Wan2.1 (14B) (Team, 2025) exceed $10^7$ dimensions. **Empirically**, we observe that guidance strategies designed for smaller models become unstable at this scale. Building on the observation by Lin et al. (2024) regarding excessive guidance at Zero-SNR, we observe that this phenomenon is further exacerbated by model scale; specifically, the guidance update

magnitude scales aggressively with the latent dimensionality. Without careful regulation, these disproportionately large updates during the initial sampling steps lead to severe visual artifacts, such as color saturation and structural incoherence, effectively causing the model to "overshoot" the realistic image distribution. This necessitates a scalable, magnitude-aware correction mechanism.

## 3. MAMBO-G: Magnitude-Aware Mitigation

### 3.1. The Risk of Zero-SNR Guidance

Rectified Flow (Liu et al., 2023) typically initializes sampling from a pure noise state $\mathbf{x}_1 \sim \mathcal{N}(\mathbf{O}, \mathbf{I})$ at time $t = 1$ to get an example $x_0$ from the original data distribution $\mathcal{X}$. The velocity field guides the noise $\mathbf{x}_1$ towards the data $\mathbf{x}_0$.

$$\mathbf{v}(\mathbf{x}_t, t, c) = \mathbb{E}_{\mathbf{x}_0 \sim \mathcal{X}, \mathbf{x}_1 \sim \mathcal{N}(\mathbf{O}, \mathbf{I})} \left[ \mathbf{x}_0 - \mathbf{x}_1 \mid \mathbf{x}_t, c \right], \quad (1)$$

where $\mathbf{x}_t$ is the intermediate state. We denote the conditional velocity as $\mathbf{v}(\mathbf{x}_t, t, c)$ and the unconditional one as $\mathbf{v}(\mathbf{x}_t, t, \varnothing)$. The classifier-free guidance substitutes $\mathbf{v}(\mathbf{x}_t, t, c)$ with $\tilde{\mathbf{v}}(\mathbf{x}_t, t, c)$ during sampling, which is defined as:

$$\begin{aligned} &\tilde{\mathbf{v}}(\mathbf{x}_t, t, c) \\ &:= w \cdot (\mathbf{v}(\mathbf{x}_t, t, c) - \mathbf{v}(\mathbf{x}_t, t, \varnothing)) + \mathbf{v}(\mathbf{x}_t, t, \varnothing), \end{aligned} \quad (2)$$

where $w$ is the guidance scale and the **guidance update** $\Delta\mathbf{v}(x_t, t, c) := \mathbf{v}(\mathbf{x}_t, t, c) - \mathbf{v}(\mathbf{x}_t, t, \varnothing)$.

At the initialization step ($t = 1$), the input $\mathbf{x}_1$ is pure noise and statistically independent of the data $\mathbf{x}_0$. Consequently, $\mathbf{x}_1$ provides no spatial or semantic cues about the target image. The model prediction for $\hat{\mathbf{x}}_0 = \mathbb{E}[\mathbf{x}_0 | \mathbf{x}_t, c]$ thus reduces to the conditional expectation that only depends on the text prompt $c$:

$$\begin{aligned} \boldsymbol{\mu}_{\text{cond}} &:= \mathbb{E}\left[\mathbf{x}_0 \mid \mathbf{x}_1, \, c\right] = \mathbb{E}\left[\mathbf{x}_0 \mid c\right], \\ \boldsymbol{\mu}_{\text{uncond}} &:= \mathbb{E}\left[\mathbf{x}_0 \mid \mathbf{x}_1, \, c\right] = \mathbb{E}\left[\mathbf{x}_0 \mid \varnothing\right]. \end{aligned} \quad (3)$$

At this initial step, the guidance update $\Delta\mathbf{v}(\mathbf{x}_1, 1, c)$ reduces to a constant offset:

$$\begin{aligned} \Delta\mathbf{v}_c &= \mathbf{v}(\mathbf{x}_1, 1, c) - \mathbf{v}(\mathbf{x}_1, 1, \varnothing) \\ &= \mathbb{E}[\mathbf{x}_0 - \mathbf{x}_1 | \mathbf{x}_1, c] - \mathbb{E}[\mathbf{x}_0 - \mathbf{x}_1 | \mathbf{x}_1, \varnothing] \\ &= (\boldsymbol{\mu}_{\text{cond}} - \mathbf{x}_1) - (\boldsymbol{\mu}_{\text{uncond}} - \mathbf{x}_1) \\ &= \boldsymbol{\mu}_{\text{cond}} - \boldsymbol{\mu}_{\text{uncond}}. \end{aligned} \quad (4)$$

This difference $\Delta\mathbf{v}$ is a **generic direction** based on dataset statistics. **It is independent of the sampled noise $\mathbf{x}_1$.**

To understand the guidance behavior at initialization, we analyze the consistency of guidance updates $\Delta\mathbf{v}$ across different random noise samples for fixed prompts. As shown in Figure 3, the cosine similarity is approximately 1.0 at

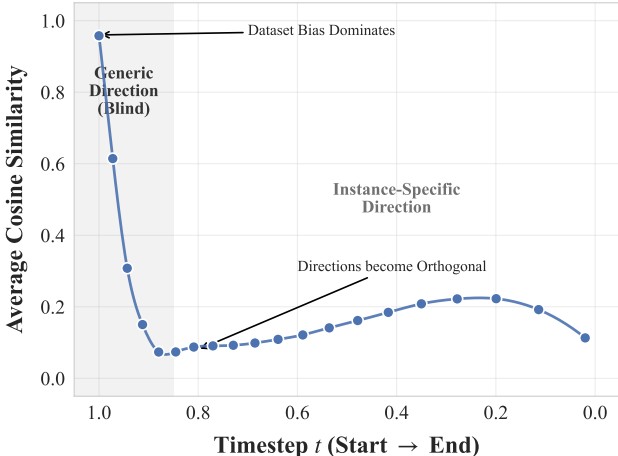

*Figure 3.* **Collapse of Guidance Directions at Initialization.** We analyze the cosine similarity of guidance updates ($\Delta\mathbf{v}$) across different noise seeds for a fixed prompt. At $t = 1.0$, similarity $\approx$ 1.0, indicating a generic direction independent of specific noise. As $t$ decreases ($t < 0.8$), updates rapidly diverge and become instance-specific. This observation motivates **MAMBO-G** to dampen the guidance scale specifically in this high-similarity, generic regime.

$t = 1$. This confirms that the initial guidance is a **generic direction** determined solely by the prompt, independent of the specific noise $\mathbf{x}_1$.

Applying a large guidance scale to this generic direction is risky. Since the update ignores the specific noise structure, a strong force can drive the trajectory away from the valid data distribution. However, this state is temporary. The similarity drops quickly as sampling continues ($t < 0.8$), and the guidance becomes **instance-specific**. This motivates **MAMBO-G** : we reduce the guidance scale during this initial generic phase to prevent instability, then boost it to further enhance the guidance effect as the image structure forms.

### 3.2. Quantifying the Relative Guidance Strength

Although the guidance direction $\Delta\mathbf{v}$ at $t = 1$ is independent of $x_1$, the magnitude of the model's conditional velocity $\mathbf{v}_{\text{cond}}$ is highly instance-dependent. To quantify the relative strength of the guidance update, we define the ratio $r_t$:

$$r_t = \frac{\|\mathbf{v}(\mathbf{x}_t, t, c) - \mathbf{v}(\mathbf{x}_t, t, \varnothing)\|_2}{\|\mathbf{v}(\mathbf{x}_t, t, \varnothing)\|_2}. \quad (5)$$

We interpret $r_t$ as an **empirically motivated heuristic for relative guidance strength**. Specifically, $r_t$ measures how large the guidance-induced discrepancy $\Delta\mathbf{v} = \mathbf{v}(\mathbf{x}_t, t, c) - \mathbf{v}(\mathbf{x}_t, t, \varnothing)$ is relative to the unconditional baseline velocity $\mathbf{v}(\mathbf{x}_t, t, \varnothing)$, thereby providing an instance-specific indicator of the relative fluctuation in the velocity field.

Intuitively, a large $r_t$ reflects strong relative fluctuations in the guidance-induced discrepancy, indicating an unstable

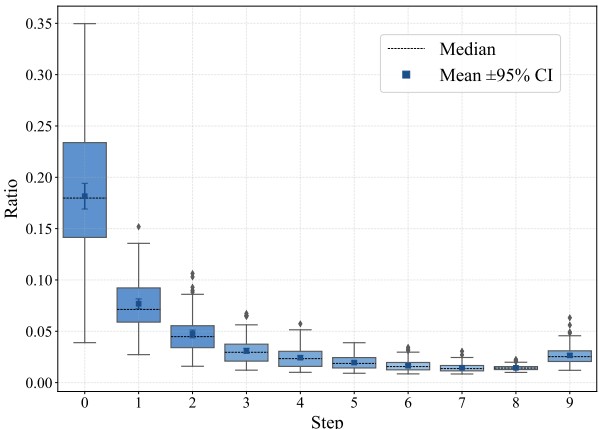

*Figure 4.* **Dynamics of the ratio during sampling.** We monitor the evolution of the relative guidance strength $r_t$ throughout the sampling process. The ratio starts at a high peak, reflecting a strong conditional influence that can lead to early-stage instability if left unregulated. It then rapidly decays and stabilizes within a few sampling steps. This empirical trend identifies the initial phase as a critical regime where guidance damping mechanism is most necessary.

conditional influence. Such excessive deviation suggests that the conflict between the prompt and the intrinsic image structure is severe, potentially leading to generation artifacts. Therefore, samples with excessively high $r_t$ are risky outliers that require mitigation.

Empirically, as shown in Figure 4, this risk is most pronounced during the initialization phase ($t \to 1$), where $r_t$ reaches its peak. This high-ratio phase coincides with the generic direction regime (Figure 3), where the guidance direction is not yet adapted to the specific noise instance. Consequently, applying a large guidance scale when $r_t$ is high risks amplifying generic, coarse features in an uncontrolled manner, rather than refining the image structure. Thus, $r_t$ serves as a robust indicator for potential instability, necessitating a damping mechanism to prevent trajectory collapse.

### 3.3. Empirical Dynamics and Sample Heterogeneity

Our observations align with Lin et al. (2024) regarding the instability of zero-SNR sampling. To further investigate this, we designed an ablation study on SD3.5. We used 100 prompts, with 20 seeds corresponding to each prompt. The ratio value at the first step was calculated for each of these 20 seeds per prompt, allowing us to divide them equally into high-ratio and low-ratio groups. Sampling was then performed using a default guidance scale of 7. The results, presented in Figure 5, demonstrate that the quality of the high-ratio group was significantly lower than that of the low-ratio group. This experimentally suggests that the ratio can serve as an effective metric for modeling sampling stability. Notably, we observe considerable variance in ratio values

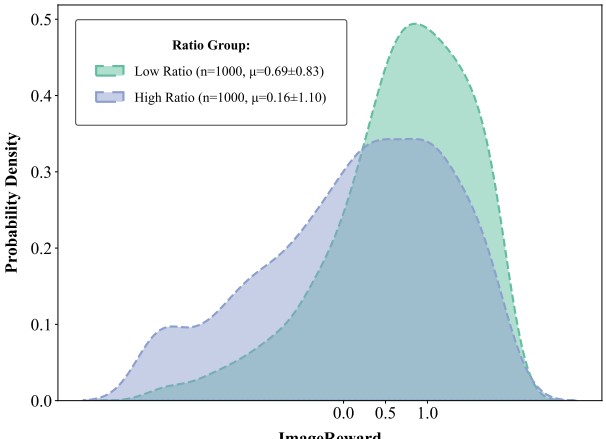

*Figure 5.* **Probability density of ImageReward scores across different Ratio groups.** We present KDE plots comparing ImageReward scores for low-ratio versus high-ratio samples at the first sampling step. The results show that lower initial ratios yield significantly higher quality, validating the ratio as a robust indicator for predicting sampling stability.

across different samples, even at the same timestep $t$. Standard dynamic guidance strategies rely on time-dependent schedules $w(t)$. However, as shown in our analysis, the ratio $r_t$ varies significantly across samples even at the same timestep. A purely time-based schedule $w(t)$ ignores this heterogeneity, failing to selectively mitigate the instability of high-ratio outliers. This observation provides empirical support for our approach of modeling guidance based on the ratio $w(r_t)$ rather than time alone.

### 3.4. MAMBO-G: Adaptive Damping Strategy

Building on the statistical insight that updates with high $r_t$ represent outliers (Eq. 5), we formulate the damping function $w(r_t)$ to mitigate the risk of these deviations. Since high-$r_t$ updates likely exceed the valid velocity distribution, relying on them with a strong guidance scale typically leads to trajectory collapse or artifacts.

To determine the optimal suppression schedule, we turn to the empirical evidence. By performing a controlled grid search for the maximum effective guidance scale across varying $r_t$ (see Algorithm 1), we derive an empirical reference curve (visualized in Figure 6). Notably, this search procedure does not assume any predefined functional form. We observe that the model's tolerance for strong guidance does not decay linearly, but drops sharply as the update becomes more disproportionate. To align the guidance strength with this empirical safe regime, we fit the stability boundary with an exponential decay function:

$$w(r_t) = 1 + (w_{\max} - 1) \cdot \exp(-\alpha r_t). \qquad (6)$$

Here, $w_{\max}$ represents the maximum allowable guidance scale, and $\alpha > 0$ is a hyperparameter calibrated to the decay rate of the reference curve.

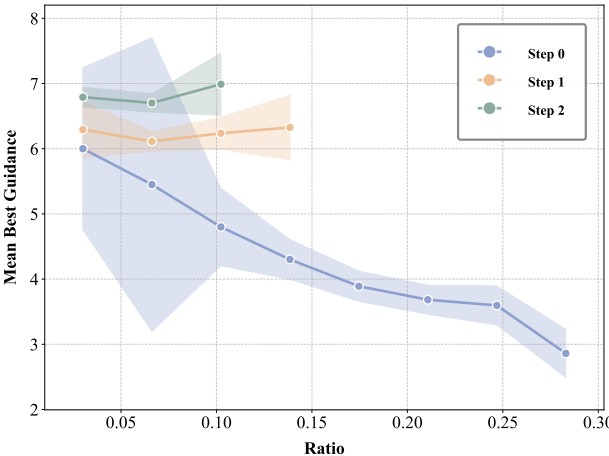

*Figure 6.* **Optimal Guidance Scale vs. Ratio.** We perform a greedy search to identify the optimal guidance scale maximizing ImageReward for various ratios. The results illustrate that the optimal scale decreases as the ratio increases, exhibiting an exponential decay. This trend directly motivates our use of an exponential damping function.

This formulation acts as a continuous, magnitude-aware filter. It permits aggressive boosting when the conditional update is statistically normal ($r_t \rightarrow 0$) but applies exponentially stronger damping as $r_t$ increases. By dynamically suppressing the specific "outlier" updates identified by $r_t$, **MAMBO-G** prevents the amplification of unstable directions while retaining the benefits of high guidance in safe regions.

## 4. Experiments

We evaluate **MAMBO-G** on image and video generation, testing its effectiveness across architectures. We also study how design choices and hyperparameters affect performance.

### 4.1. Text-to-Image Generation

We test **MAMBO-G** on text-to-image generation using two recent models: Stable Diffusion v3.5 (SD3.5) and Lumina-Next (see Figure 7). We measure quality with ImageReward and CLIPScore, comparing against the base samplers. Across both models, **MAMBO-G** improves image quality and semantic alignment. With fewer sampling steps, **MAMBO-G** matches or exceeds longer-step baselines, speeding up generation without changing the underlying sampler. We measure its improvement by FID as well (see Section B.1).

### 4.2. Text-to-Video Generation

We apply **MAMBO-G** to video diffusion models and evaluate on vBench metrics for visual quality and aesthetics (see

Figure 8). Compared with the original guidance schedules, models with **MAMBO-G** produce videos with higher quality and better semantic alignment at the same step count. In some cases, a smaller backbone with **MAMBO-G** matches the quality of a larger baseline.

### 4.3. Impact of Dimensionality

*Table 1.* **Quantitative results of text-to-image generation across different resolutions measured by ImageReward. MAMBO-G** demonstrates consistent performance, whereas CFG degrades significantly at higher resolutions.

| Resolution | CFG (Baseline) | MAMBO-G (Ours) |
|---|---|---|
| $256 \times 256$ | 0.53 | **0.83** |
| $512 \times 512$ | 0.63 | **1.10** |
| $768 \times 768$ | 0.30 | **1.07** |
| $1024 \times 1024$ | 0.20 | **1.02** |

To verify that guidance instability increases with dimensionality, we evaluate **MAMBO-G** across different resolutions using Qwen-Image, ranging from $256 \times 256$ to $1024 \times 1024$.

We compare the ImageReward of the baseline guidance against **MAMBO-G** . As shown in Table 1, the performance gap widens significantly as resolution increases. At lower resolutions, the baseline performs adequately, and the gain from **MAMBO-G** is marginal. However, at $1024 \times 1024$, the baseline frequently suffers from over-saturation and artifacts, whereas **MAMBO-G** maintains structural coherence (see Figure 2). This trend confirms that the risk of unscaled guidance updates is inherently linked to the total noise magnitude in high-dimensional spaces, making our method particularly vital for future high-resolution video models.

### 4.4. Compatibility with Advanced Guidance Strategies

*Table 2.* **Orthogonality analysis of MAMBO-G with other methods.** The results show that **MAMBO-G** can be seamlessly integrated with other methods like APG (Sadat et al., 2024) and Rescale (Lin et al., 2024) to further improve performance.

| Method | ImageReward |
|---|---|
| Baseline (Constant CFG) | 0.12 |
| Rescale | 0.73 |
| Rescale + **MAMBO-G** | **1.12** |
| APG | 0.85 |
| APG + **MAMBO-G** | **0.96** |

A key advantage of **MAMBO-G** is its orthogonality to other guidance optimization techniques. Since our method exclusively modulates the guidance scale, it can be directly stacked with methods that normalize the update vector or alter the sampling trajectory.

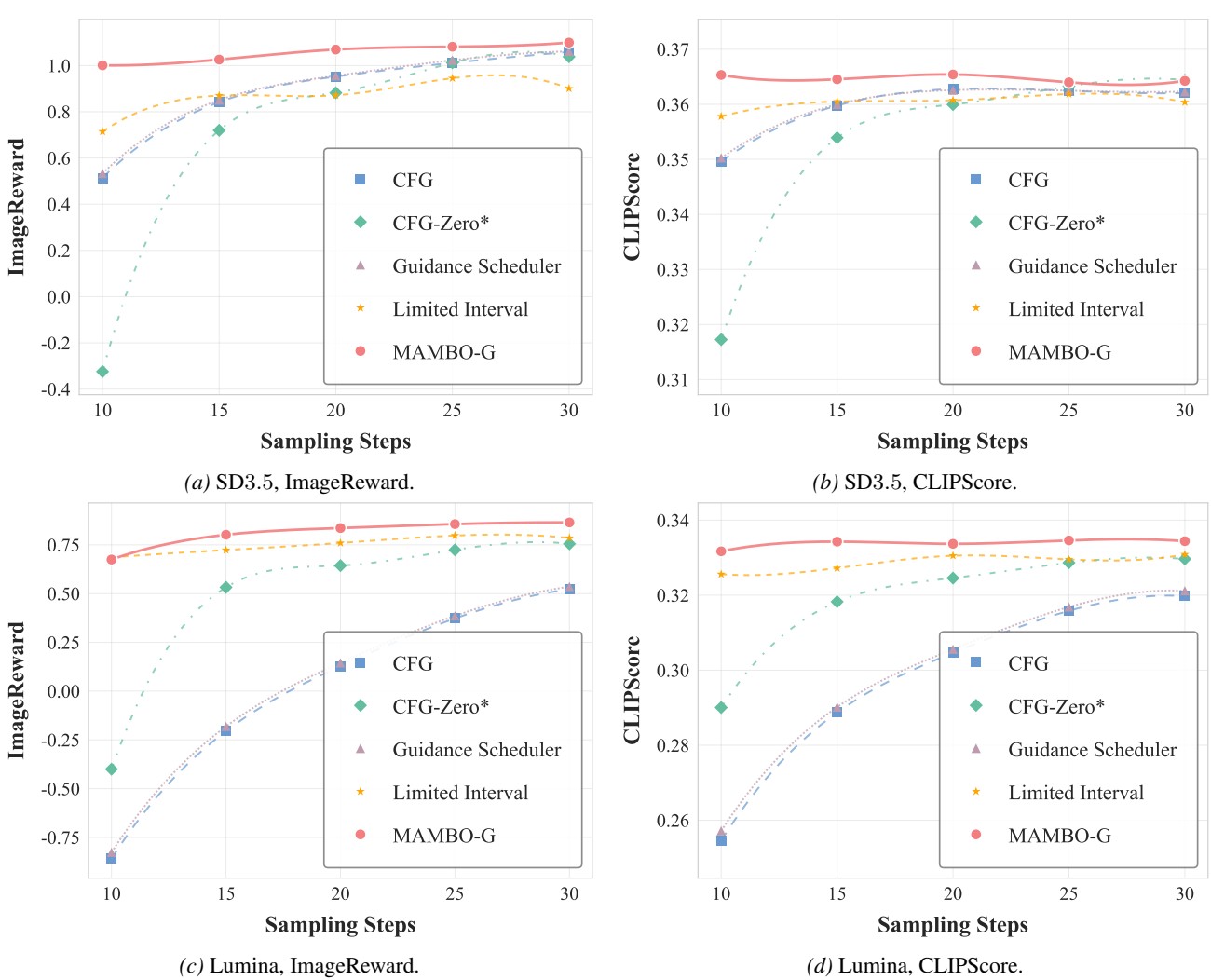

*(a)* SD3.5, ImageReward.

*(b)* SD3.5, CLIPScore.

*(c)* Lumina, ImageReward.

*(d)* Lumina, CLIPScore.

*Figure 7.* **Quantitative results of comparative analysis with other baselines in text-to-image generation measured by ImageReward and CLIPScore.** Here, **Guidance Scheduler** refers to Xi et al. and **Limited Interval** refers to Kynkäänniemi et al. (2024). The results demonstrate **MAMBO-G**'s remarkable superiority over others in low-step generation, achieving the quality of 30-step generation of CFG in only **10 steps.**

To verify this, we evaluate **MAMBO-G** in combination with Guidance Rescale (GR) (Lin et al., 2024) and Adaptive Projection Guidance (APG) using the Qwen-Image model. GR rescales the guidance vector to prevent over-exposure, while APG dynamically adjusts the projection direction. As shown in our comparisons (Table 2), while both baselines effectively mitigate some artifacts, stacking them with **MAMBO-G** yields further consistent improvements in ImageReward. This plug-and-play nature ensures **MAMBO-G** remains relevant even as new guidance strategies are developed.

### 4.5. Ablation Studies and Hyperparameters

We perform ablation studies to understand the impact of our modeling choices.

**Guidance schedule.** We compare several ways of mapping

$r_t$ to a guidance scale, including exponential decay, linear decay, and simple inverse functions. Exponential decay provides a good balance between stability and detail in our experiments (see Figure 9), so we use it as the default.

*Table 3.* **Comparison with time-based schedule on Qwen-Image (10 steps).** The time-based schedule uses the average guidance scale of **MAMBO-G** at each step. The results highlight the importance of instance-level adaptation.

| Method | ImageReward |
|---|---|
| Baseline (Constant CFG) | 0.12 |
| Time-based Schedule | 0.83 |
| **MAMBO-G** (Ours) | **1.08** |

**Instance-aware vs. Time-based Schedule.** A natural ques-

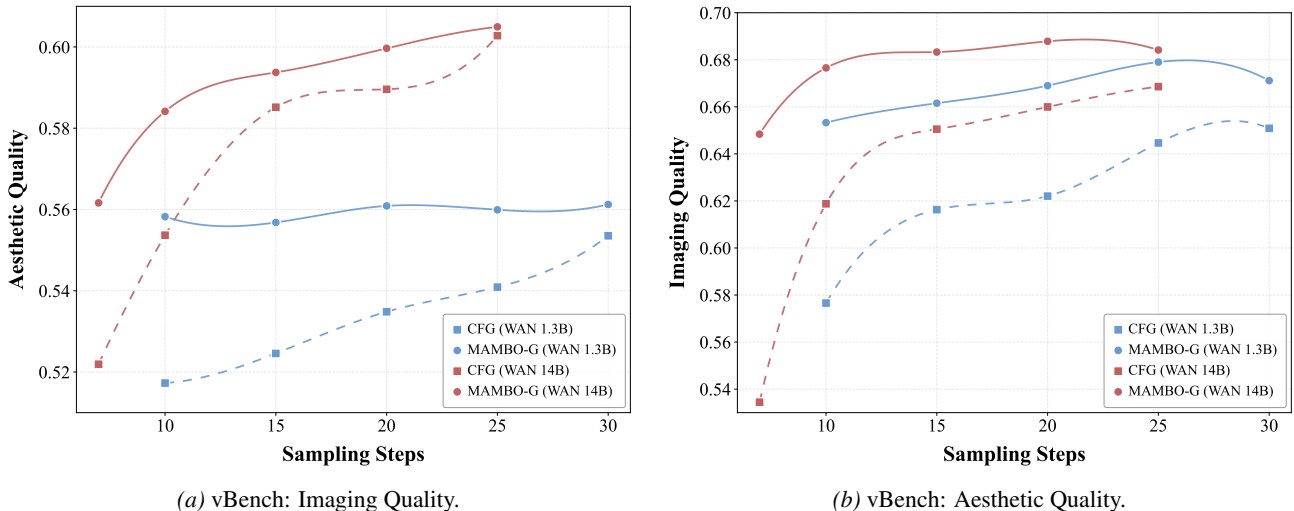

*(a)* vBench: Imaging Quality.

*(b)* vBench: Aesthetic Quality.

*Figure 8.* **Quantitative results of text-to-video generation, comparing CFG and MAMBO-G under vBench.** The results strongly validate the effectiveness of **MAMBO-G** on video generations, even overtaking CFG-Wan 14B just on Wan 1.3B.

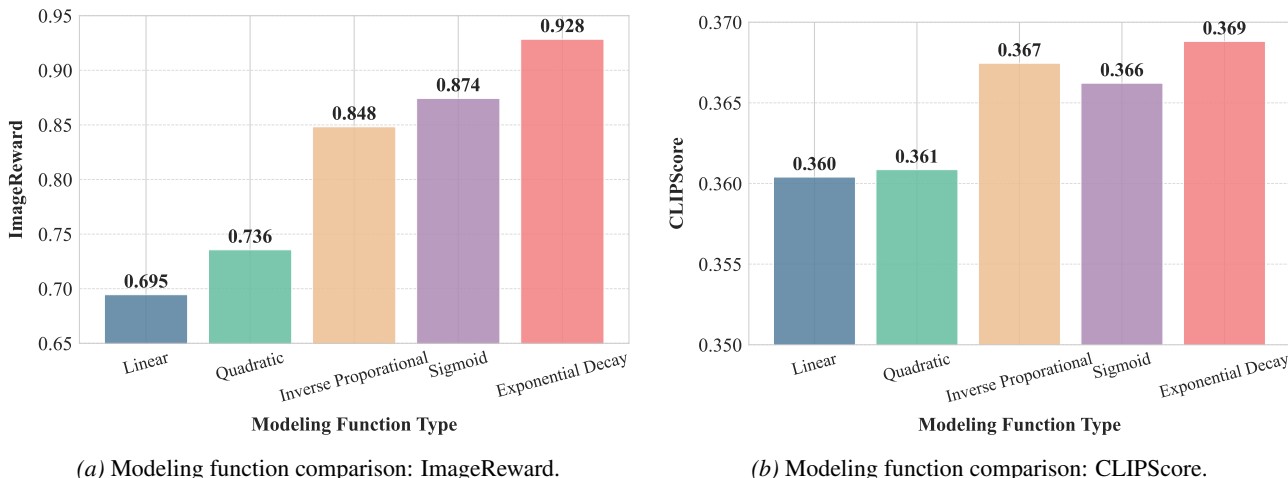

*(a)* Modeling function comparison: ImageReward.

*(b)* Modeling function comparison: CLIPScore.

*Figure 9.* **Ablation studies on different types of modeling functions.** The comparative results demonstrate that the exponential decay function delivers the optimal fitting results among other functions, validating the soundness of our method.

tion is whether a simple time-dependent schedule suffices. To investigate this, we constructed a "Time-based Schedule" baseline by averaging the effective guidance scale $w(r_t)$ of **MAMBO-G** across all samples at each timestep, then applying this fixed curve to all generated images. As shown in Table 3, the time-based schedule significantly improves over the constant baseline (ImageReward $0.12 \rightarrow 0.83$), confirming that dampening early-stage guidance is generally beneficial. However, **MAMBO-G** achieves a further substantial improvement ($0.83 \rightarrow 1.08$). This gap underscores the critical role of *instance-awareness*: since instability varies across random seeds, a fixed schedule over-penalizes stable samples or under-penalizes risky ones, whereas **MAMBO-G** adapts dynamically.

**Hyperparameter sensitivity.** We vary $w_{\max}$ and the decay

rate $\alpha$ over a range of values (see Tables 4 and 5). **MAMBO-G** maintains stable behavior and competitive scores across a broad region, suggesting it does not require heavy tuning. These results support our default setting as a robust compromise rather than a narrowly tuned optimum.

**Scheduler generalization.** We also apply **MAMBO-G** on ODE solvers such as UniPC (see Figure 10). The method still improves over the corresponding baselines, showing it can be combined with different samplers.

## 5. Conclusion

In this work, we address the instability of Classifier-Free Guidance (CFG) in large-scale models by identifying risks from excessive magnitudes during initialization. We pro-

*Table 4.* Ablation studies on hyperparameter $w_{\max}$ (with fixed $\alpha = 8$) across different models. The scores represent ImageReward, with the best performance in each row highlighted in **bold**.

| Model | Method | $w_{\max}$ | | | | | |
|---|---|---|---|---|---|---|---|
| | | **6** | **8** | **10** | **12** | **14** | **16** |
| SD3.5 | **MAMBO-G** | 0.755 | 0.830 | 0.840 | **0.878** | 0.833 | 0.819 |
| | **MAMBO-G** + Rescale | 0.740 | 0.838 | 0.905 | 0.895 | 0.903 | **0.912** |
| Qwen-Image | **MAMBO-G** | **1.112** | 1.094 | 1.081 | 0.946 | 0.899 | 0.780 |
| | **MAMBO-G** + Rescale | **1.126** | 1.095 | 1.122 | 1.093 | 1.058 | 1.042 |

*Table 5.* Ablation studies on hyperparameter $\alpha$ (with fixed $w_{\max} = 10$) across different models. The scores represent ImageReward, with the best performance in each row highlighted in **bold**.

| Model | Method | $\alpha$ | | | | | |
|---|---|---|---|---|---|---|---|
| | | **6** | **8** | **10** | **12** | **14** | **16** |
| SD3.5 | **MAMBO-G** | 0.827 | 0.840 | **0.857** | 0.817 | 0.785 | 0.774 |
| | **MAMBO-G** + Rescale | 0.875 | **0.905** | 0.867 | 0.818 | 0.808 | 0.768 |
| Qwen-Image | **MAMBO-G** | 0.830 | 1.081 | 1.156 | **1.161** | 1.132 | 1.122 |
| | **MAMBO-G** + Rescale | 1.010 | 1.122 | **1.155** | 1.142 | 1.136 | 1.130 |

pose **MAMBO-G** , a training-free strategy that dynamically modulates the guidance scale based on the update-to-base ratio $r_t$. Unlike fixed scales, **MAMBO-G** adaptively prevents artifacts during critical early steps, ensuring realistic generation trajectories. Experiments on text-to-image and text-to-video tasks demonstrate that **MAMBO-G** significantly accelerates inference and stabilizes high-resolution generation without architectural changes. Our results highlight magnitude-aware control as a robust, efficient component for state-of-the-art foundation models.

**Limitations.** We note two limitations of **MAMBO-G** . **(1) Guidance direction is not corrected: MAMBO-G** can effectively modulate guidance magnitude but does not alter guidance direction, and therefore may attenuate a semantically misleading guidance signal without fully correcting it. **(2) Lightweight hyperparameter tuning:** The choice of $w_{\max}$ still depends mildly on the model's native CFG scale, although our results suggest that **MAMBO-G** remains robust within a reasonably broad range.

## Acknowledgements

This work benefited greatly from the support of all co-authors. Fan Cheng and Ruili Feng provided overall guidance and supplied the computational resources. Shang-wen Zhu was deeply involved throughout the entire project. Qianyu Peng and Zhilei Shu helped design the experimental coding pipeline. Yuting Hu and Han Zhang contributed valuable theoretical insights. Andy Zheng offered insightful advice on manuscript writing and experimental design. Xinyu Cui and Jian Zhao also proposed constructive suggestions for the paper. This work was also supported in part by the National Key R&D Program of China under Grant 2022YFA1005000, and in part by the NSFC under Grant 61701304.

## Impact Statement

This paper presents work whose goal is to advance the field of machine learning, specifically focusing on **optimizing conditional generation**. Our proposed methods contribute to the broader scientific community by **improving the efficiency and coherence of conditional generative models**, which has potential applications in **text-to-image and text-to-video generation**.

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

# A. Experimental Settings

In this section, we provide a comprehensive description of the experimental configurations to ensure the reproducibility of our results. We detail the model specifications, sampling schedulers, datasets, and randomization protocols employed in our evaluations.

## A.1. Models and Schedulers

We conduct experiments across a diverse set of generative models. The specific sampling schedulers and hyperparameter configurations for each model are detailed below:

- **Stable Diffusion v**3.5: We employ the Flow Matching Euler scheduler by default, while the UniPC scheduler is employed for ablation studies. For the Classifier-Free Guidance (CFG) baseline, a guidance scale of 7.0 is applied.

- **Lumina-Next**: Inference is performed using the UniPC scheduler. The guidance scale is set to 7.0 for the CFG baseline.

- **Wan**2.1: Similarly, this model uses the UniPC scheduler; however, the guidance scale is adjusted to 5.0 for the CFG baseline to ensure optimal performance in video generation.

- **Wan**2.2: This model uses Flow Matching Euler scheduler; the guidance scale is adjusted to 5.0 for the CFG baseline to ensure optimal performance in video generation.

- **Qwen-Image**: This model uses Flow Matching Euler scheduler; the guidance scale is adjusted to 4.0 for the CFG baseline to ensure optimal performance in video generation.

## A.2. Datasets and Randomization

To guarantee reliability and deterministic generation, we specify the datasets and seed strategies used for quantitative evaluation in the main paper:

- **Text-to-Image Generation (Figure 7)**: Evaluations are performed on the **MS-COCO**. We construct a test set comprising the initial 500 prompts extracted from the full dataset.

  - **Randomization**: To ensure reproducibility, we adopt a deterministic strategy where the random seed for each sample is set equal to its corresponding prompt index.

- **Text-to-Video Generation (Figure 8)**: Video synthesis capabilities are evaluated using **WebVid**. For this task, we randomly select a subset of 100 samples.

  - **Randomization**: The randomization protocol remains consistent with the text-to-image setting (i.e., seed equals sample index).

- **Impact of Dimensionality (Table 1)**: A specific subset consisting of the first 200 prompts from **ImageReward Dataset (Xu et al., 2023)** is employed for this analysis.

  - **Randomization**: To maintain fixed noise initialization across experiments, the random seed is assigned to match the prompt ID of each sample.

- **Orthogonality Analysis (Table 2)**: We use a curated collection of 200 prompts drawn from **ImageReward Dataset**.

  - **Randomization**: Stochasticity is governed by a deterministic mapping, where the seed for each generation is aligned with the prompt's index in the test suite.

- **Comparison with Time-Based Schedule (Table 3)**: Testing is conducted using the first 200 samples extracted from **ImageReward Dataset**.

  - **Randomization**: We adhere to a predefined scheme in which the random seed for each instance is set equal to its sequence number in the group.

## A.3. Hyper-Parameter Settings

For all experiments involving **MAMBO-G** , we use the default parameters $\alpha = 8$ and $w_{\max} = 10$, which provide robust results across diverse scenarios.

In addition to **MAMBO-G** , we detail the configurations for all baselines discussed in Section 4.1 as follows:

- **CFG-Zero\* (Fan et al., 2025b):** This method is parameter-free. We strictly follow the implementation details provided in the original work.

- **Guidance Scheduler (Xi et al.):** We employ a cosine schedule for guidance and set the minimum guidance scale to 4.0.

- **Limited Interval (Kynkäänniemi et al., 2024):** CFG is applied during the interval spanning $10\%$ to $90\%$ of the total sampling steps (i.e., excluding the first and last $10\%$). The guidance scale is fixed at 7.0.

## A.4. Prompts for Visual Examples

In this subsection, we list the exact textual prompts corresponding to the visual qualitative results presented in the main paper.

**Figure 1 (a):** *"Cirno, Touhou, ice fairy, light blue short hair, big blue hair ribbon, blue eyes, smug face, open mouth, confidence, blue and white dress, serrated skirt, red neckerchief, crystal wings, ice wings, floating, hands on hips, ice magic, snowflakes, frozen lake background, misty, magical atmosphere, anime style, cel shading, masterpiece, best quality, 8k, vivid colors"*

**Figure 1 (b):** *"A photograph of an astronaut riding a horse, high quality, 4k, detailed, on Moon"*

**Figure 1 (c) (d):** *"Two anthropomorphic cats in comfy boxing gear and bright gloves fight intensely on a spotlighted stage."*

**Figure 2:** *"delicious plate of food"*

## A.5. Configurations of Exploratory Experiments

Here we specify the detailed settings for the exploratory analyses discussed in the main text.

- **Collapse of Guidance Directions at Initialization (Figure 3):** These results are derived from Stable Diffusion v3.5 using a default guidance scale of 7.0. We configure the random noise seeds and use the fixed prompts from MS-COCO.

- **Dynamics of the ratio during sampling (Figure 4):** Ratio values are tracked using Stable Diffusion v3.5 with a guidance scale of 7.0. The sampling process consists of 10 steps. The dataset comprises 100 prompts from MS-COCO, following the randomization strategy defined in Section 4.1.

- **Probability density of ImageReward scores across different ratio groups (Figure 5):** Using Stable Diffusion v3.5 (guidance scale 7.0), we analyze the first 20 prompts from MS-COCO. We employ a 10-step sampling procedure for each generation. For each prompt, we generate samples across 20 distinct seeds. Ratio groups are determined via binary splitting based on the median ratio value.

- **Optimal Guidance Scale vs. Ratio (Figure 6):** We search for optimal guidance scales relative to observed ratio values on Stable Diffusion v3.5. A constant guidance schedule of 7.0 serves as the baseline. The generation process uses 10 sampling steps. For the first step, we sweep the guidance scale from 1.5 to 9.0 in increments of 0.5 to identify the optimal configuration, then searching for the subsequent step based on the previously optimal guidance schedule. Detailed algorithm is presented in Algorithm 1. This experiment uses prompts from MS-COCO, adhering to the randomization strategy described in Section 4.1.

---

**Algorithm 1** Greedy Search for Step-wise Optimal Guidance Scale

---

**Require:** Prompts $\mathcal{P}$ from MS-COCO, Sampling steps $T = 10$, Search space $\mathcal{W} = \{1.5, 2.0, \ldots, 9.0\}$, Baseline scale $w_{base} = 7.0$

**Ensure:** Optimal guidance schedule $\mathbf{w}^* = (w_1^*, w_2^*, \ldots, w_T^*)$

1: Initialize $\mathbf{w}^* = (w_{base}, w_{base}, \ldots, w_{base})$ {Set baseline schedule}
2: **for** $t = 1$ **to** $T$ **do**
3:     {Search for the optimal scale at current step $t$}
4:     $w_t^* = \arg\max_{w \in \mathcal{W}}$ AverageMetric (Generate($\mathcal{P}, \mathbf{w}_{tmp}$))
5:     **where** $\mathbf{w}_{tmp} = (w_1^*, \ldots, w_{t-1}^*, w, w_{base}, \ldots, w_{base})$
6:     Update $\mathbf{w}^*$ with the newly found $w_t^*$
7: **end for**
8: **Output:** Correlate each $w_t^*$ with the corresponding average ratio $r_t$ observed at step $t$ to analyze the trend in Figure 6.
9: **return** $\mathbf{w}^*$

---

# B. Supplementary Results

## B.1. Quantitative Evaluation via FID

To rigorously assess the distributional similarity between generated images and real-world data, we evaluate **MAMBO-G** using the Fréchet Inception Distance (FID) on 5,000 prompts from MS-COCO dataset. All samples are generated under the fixed seed. The results, summarized in Table 6, demonstrate that **MAMBO-G** significantly enhances the generative quality in low-step regimes. Specifically, at only 10 sampling steps, **MAMBO-G** achieves an FID of **32.0**, representing a substantial improvement over the standard CFG baseline (63.6). Notably, our 10-step performance closely approaches the quality of 30-step CFG (24.8), effectively bridging the gap between efficient sampling and high-fidelity generation. This confirms that **MAMBO-G** still maintains an accurate generation trajectory even under aggressive acceleration.

*Table 6.* **FID Results on MS-COCO.** We compare **MAMBO-G** with standard CFG across different sampling steps. FID ($\downarrow$) measures the distributional distance to reference images (we use 50-step CFG as reference). The results demonstrate that **MAMBO-G** at 10 steps significantly outperforms the 10-step CFG baseline.

| Method | Sampling Steps | FID ($\downarrow$) |
|---|---|---|
| CFG (Baseline) | 10 | 63.6 |
| CFG (Baseline) | 30 | 24.8 |
| **MAMBO-G (Ours)** | **10** | **32.0** |
| CFG (Reference) | 50 | — |

We further compare **MAMBO-G** against representative dynamic-guidance baselines in terms of FID to provide a broader distributional evaluation. We evaluate all methods on 1,000 prompts from MS-COCO using Qwen-Image at 10 sampling steps, with 50-step CFG serving as the reference distribution. As shown in Table 7, **MAMBO-G** achieves the lowest FID among all 10-step methods, outperforming Guidance Scheduler (Xi et al.), CFG-Zero* (Fan et al., 2025b), and Limited Interval (Kynkäänniemi et al., 2024) by a substantial margin. This result is consistent with the ImageReward-based comparisons in the main text and further confirms that **MAMBO-G** produces generation trajectories that are globally closer to the high-fidelity reference distribution.

*Table 7.* **FID comparison with dynamic-guidance baselines on MS-COCO.** All methods use Qwen-Image at 10 sampling steps. FID ($\downarrow$) is computed against a 50-step CFG reference distribution on 1,000 prompts.

| Method | FID ($\downarrow$) |
|---|---|
| CFG (Baseline, 10 steps) | 63.6 |
| Guidance Scheduler (Xi et al.) | 55.7 |
| CFG-Zero* (Fan et al., 2025b) | 46.9 |
| Limited Interval (Kynkäänniemi et al., 2024) | 38.0 |
| **MAMBO-G (Ours)** | **32.0** |

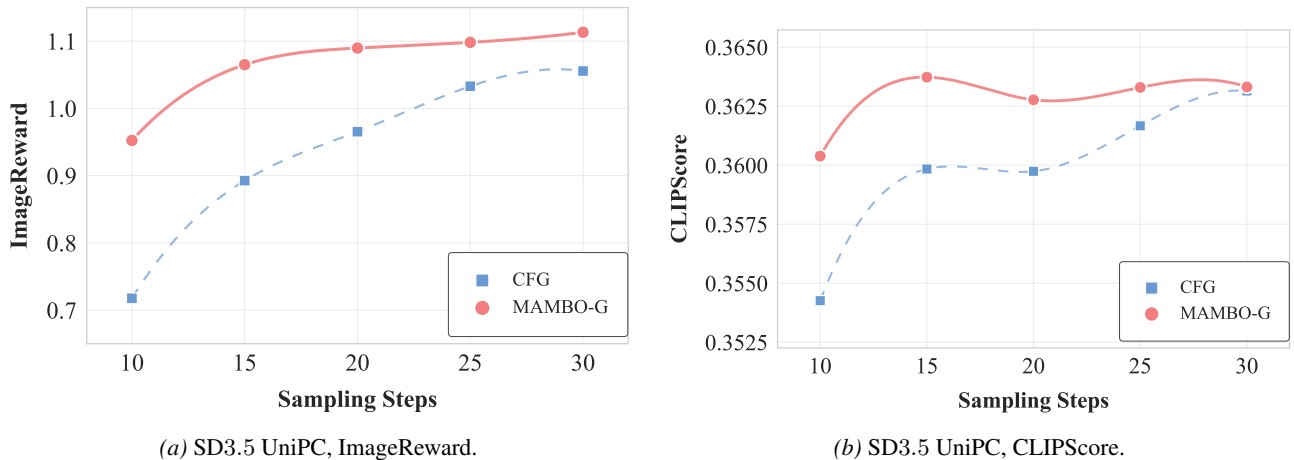

*(a)* SD3.5 UniPC, ImageReward.  *(b)* SD3.5 UniPC, CLIPScore.

*Figure 10.* Ablation studies of schedulers on UniPC. The comparative results present the consistent superiority of **MAMBO-G** over CFG across different schedulers, further validating the wide-ranging adaptability of our method.

## B.2. Experimental Details for Ablation Studies

In this section, we provide detailed configurations for the ablation studies reported in Section 4.5. The quantitative results for $w_{\max}$ and $\alpha$ are presented in the main text, while additional scheduler ablation results are shown in Figure 10. The detailed configurations are as follows:

- **Ablation studies on $w_{\max}$ (Table 4):** We evaluate the sensitivity of $w_{\max}$ across SD3.5 and Qwen-Image models. Experiments are conducted on the first 100 prompts from MS-COCO, following the randomization strategy defined in Section 4.1. The hyperparameter $\alpha$ is fixed at 8, while $w_{\max}$ is varied from 6 to 16. Each generation is performed with 10 sampling steps.

- **Ablation studies on $\alpha$ (Table 5):** We evaluate the sensitivity of $\alpha$ across SD3.5 and Qwen-Image models. Experiments are conducted on the first 100 prompts from MS-COCO, following the randomization strategy defined in Section 4.1. The hyperparameter $w_{\max}$ is fixed at 10, while $\alpha$ is varied from 6 to 16 to observe its impact on guidance damping. Each generation is performed with 10 sampling steps.

- **Ablation studies on schedulers (Figure 10):** We evaluate the robustness of **MAMBO-G** on the UniPC scheduler. Experiments are conducted on the first 100 prompts from MS-COCO, following the randomization strategy outlined in Section 4.1. The hyperparameter settings remain consistent with those described in Section 4.1. Each generation is performed with 10 sampling steps.

## B.3. Overshooting Analysis under Varying Guidance Scales and Step Sizes

We present a quantitative analysis of trajectory deviation to characterize the conditions under which early-step guidance becomes destabilizing. Specifically, we measure the $\ell_2$ distance between the intermediate state $\mathbf{x}_t$ at $t \approx 0.9$ and the corresponding state on an 80-step, $w = 4.0$ reference trajectory, which we treat as the ideal generation path. We evaluate Qwen-Image at $1024 \times 1024$ resolution under two step-count regimes: 60 steps (small $\Delta t$) and 10 steps (large $\Delta t$), sweeping the guidance scale $w$ from 1.0 to 10.0.

As shown in Table 8, the high-NFE setting remains near the reference trajectory across all guidance scales, reflecting that small step sizes afford the model sufficient opportunity to self-correct. In contrast, the low-NFE setting shows a sharp monotonic increase in deviation as $w$ grows beyond 4.0, confirming that the generic guidance direction at initialization poses a practical risk precisely when large step sizes are used. **MAMBO-G** addresses this regime by dynamically suppressing $w$ through the instance-level ratio $r_t$, thereby preventing the trajectory from deviating into unstable regions during the critical early steps.

*Table 8.* **Trajectory deviation at $t \approx 0.9$ under different step counts and guidance scales.** Deviation is measured as the $\ell_2$ distance to an 80-step, $w = 4.0$ reference trajectory on Qwen-Image ($1024 \times 1024$). A high NFE setting (small $\Delta t$) remains stable across guidance scales, whereas a low NFE setting (large $\Delta t$) exhibits rapidly growing deviation at high guidance scales, confirming that overshooting is jointly determined by step size and guidance magnitude.

| Setting | $w = 1.0$ | $w = 2.0$ | $w = 4.0$ | $w = 6.0$ | $w = 8.0$ | $w = 10.0$ |
|---|---|---|---|---|---|---|
| High NFE (60 steps, small $\Delta t$) | 0.035 | 0.022 | 0.009 | 0.017 | 0.027 | 0.037 |
| Low NFE (10 steps, large $\Delta t$) | 0.035 | 0.028 | 0.052 | 0.091 | 0.133 | 0.175 |

### B.4. Dimension-Invariance of the Magnitude-Aware Ratio

We analyze whether the ratio $r_t$ remains stable as the spatial dimensionality of the latent space increases. Under standard Gaussian initialization, both the numerator $\|\Delta \mathbf{v}\|_2$ and the denominator $\|\mathbf{v}_{\text{uncond}}\|_2$ scale proportionally with $\sqrt{d}$, where $d$ is the latent dimension. Consequently, their ratio is expected to be dimension-invariant in expectation.

To validate this empirically, we evaluate $r_t$ at the first sampling step ($t = 1$) on Qwen-Image across four resolutions from $256 \times 256$ to $1024 \times 1024$, using the same prompts and seeds.

*Table 9.* **Magnitude-aware ratio $r_t$ at the first sampling step across resolutions.** We report the mean $r_t$ at $t = 1$ on Qwen-Image under identical prompts and seeds. The ratio remains stable across a $16\times$ increase in latent dimensionality, confirming that $r_t$ is dimension-invariant and thus provides a resolution-agnostic instability signal.

| Resolution | $256 \times 256$ | $512 \times 512$ | $768 \times 768$ | $1024 \times 1024$ |
|---|---|---|---|---|
| $r_t$ ($t = 1$) | 0.255 | 0.270 | 0.259 | 0.260 |

As shown in Table 9, the value of $r_t$ remains in the range $[0.255, 0.270]$ across all four resolutions despite a $16\times$ increase in the number of latent dimensions. This confirms that the ratio scales consistently with dimensionality and that the same default hyperparameters can be applied across models operating at different resolutions without recalibration.

### B.5. Cross-Model Validation of the Exponential Decay Trend

The greedy search presented in Section A.5 identifies the empirically optimal guidance scale as a function of $r_t$ on SD3.5 and MS-COCO. To examine whether the resulting exponential trend generalizes beyond this specific setup, we conduct the same fitting procedure on Qwen-Image using two independent prompt sources: the ImageReward benchmark (Xu et al., 2023) and the MS-COCO validation captions. For each source, we use 500 prompts and 10 sampling steps.

*Table 10.* **Goodness-of-fit ($R^2$) of the exponential decay model across prompt sources and models.** We fit the empirically optimal guidance scale curve $w^*(r_t)$ to an exponential decay function and report the coefficient of determination $R^2$. Results are obtained on Qwen-Image with 500 prompts and 10 sampling steps per source.

| Model | Prompt Source | $R^2$ |
|---|---|---|
| Qwen-Image | ImageReward Benchmark | 0.92 |
| Qwen-Image | MS-COCO Val Captions | 0.94 |

As reported in Table 10, the exponential decay model achieves $R^2$ values of 0.92 and 0.94 on the two prompt sources respectively, demonstrating that the decreasing relationship between optimal guidance scale and $r_t$ is consistent across different datasets and is not an artifact of the particular SD3.5/MS-COCO combination used in the main analysis. These results further support the use of the exponential form as the default damping schedule.

The cross-model consistency of the exponential decay trend further motivates a practical transfer heuristic for applying **MAMBO-G** to unseen architectures. For practitioners porting **MAMBO-G** to a previously unseen model, we recommend setting $w_{\text{max}}$ based on the model's native optimal CFG scale, which is typically reported in the official model card and requires no additional search, and fixing the decay rate to $\alpha = 12$ as a conservative universal default. Under this configuration, the only architecture-specific input is $w_{\text{max}}$, while $\alpha$ can be treated as a constant across models. This recommendation is supported by our hyperparameter ablation studies (Section B.2), which demonstrate that **MAMBO-G** maintains stable performance over a broad range of both $\alpha$ and $w_{\text{max}}$ values, confirming that fine-grained per-model tuning is not required when adapting the method to a new architecture.

## B.6. Runtime Overhead

We report wall-clock timing measurements to verify that the NFE reduction achieved by **MAMBO-G** translates directly into practical speedup. The only additional computation introduced by **MAMBO-G** is a global $\ell_2$ norm over the flattened latent tensor, which has complexity $O(D)$ — negligible relative to the $O(D^2)$ self-attention in the DiT backbone. We measure per-step latency on an H100 GPU for Qwen-Image at $512 \times 512$ and for Wan2.2-I2V-A14B.

*Table 11.* **Wall-clock overhead of MAMBO-G per denoising step.** DiT backbone latency and the additional norm computation time are reported on an H100 GPU. The overhead introduced by **MAMBO-G** is invariant to batch size and constitutes less than 0.02% of the total per-step cost across all settings.

| Setting | DiT backbone | MAMBO-G norm | Overhead |
|---|---|---|---|
| Qwen-Image ($512^2$), bs $= 1$ | 314.8 ms | 0.045 ms | 0.014% |
| Qwen-Image ($512^2$), bs $= 4$ | 461.3 ms | 0.045 ms | <0.01% |
| Wan2.2-I2V-A14B, bs $= 1$ | 30,762 ms | 0.39 ms | 0.001% |

As shown in Table 11, the norm computation adds at most 0.045 ms per step regardless of batch size, amounting to less than 0.02% of total per-step latency across all evaluated configurations. For Wan2.2-I2V-A14B, where each denoising step takes over 30 seconds, the overhead is effectively zero. These results confirm that any wall-clock speedup from **MAMBO-G** is fully attributable to the reduction in NFE rather than an increase in per-step cost.

