# OpenReview forum: "MAMBO-G: Magnitude-Aware Mitigation for Boosted Guidance"
_ICML.cc/2026/Conference — ICML 2026 regular_

### Official Review · Reviewer_LPY1 · 2026-02-14

**Soundness:** 2
**Presentation:** 3
**Significance:** 2
**Originality:** 2
**Overall Recommendation:** 4
**Confidence:** 3

**Summary:**

This paper introduces MAMBO-G, a training-free acceleration framework that tries to reduce computational cost by dynamically optimizing guidance magnitudes. The core idea is to explicitly model magnitude variations—often arising from covariate shifts, scale drift, or non-stationarity—through a magnitude-aware normalization and masking strategy integrated into a masked autoencoding paradigm.

**Compliance With Llm Reviewing Policy:**

Affirmed.

**Final Justification:**

My concerns have been resolved, and I maintain my rating.

**Key Questions For Authors:**

1. Some distribution shifts are synthetically constructed; it remains unclear how representative they are of real-world deployment scenarios.
2. Although ablations are provided, further disentanglement between masking strategy, normalization design, and loss formulation would clarify which component drives the majority of gains.

**Limitations:**

1. Its reliance on normalization-based correction may introduce sensitivity to extreme outliers or heavy-tailed distributions.

**Strengths And Weaknesses:**

Strength:
1. The magnitude-aware normalization mechanism is conceptually simple yet effective, enabling the model to preserve scale information rather than discarding it through aggressive normalization.
2. The authors evaluate under multiple shift scenarios (e.g., scale amplification, regime changes), providing both quantitative and qualitative analyses.

Weakness:
1. While well-executed, the magnitude-aware normalization can be viewed as a relatively modest extension of existing normalization and scale-adaptive techniques.
2. The paper motivates magnitude awareness intuitively, but lacks a formal analysis of why the proposed mechanism improves robustness under specific shift assumptions.

---

> ### Author Rebuttal · Authors · 2026-03-29
>
> ## Response to Reviewer LPY1
>
> We sincerely thank Reviewer LPY1 for the time and effort invested in reviewing our work, and for the positive Weak Accept recommendation.
>
> **Clarification of method design**
> We would like to take this opportunity to provide a brief clarification of our method's design, as some of the terminology in the review may reflect a slight mismatch with our paper's framing. To ensure the reviewer has the most accurate picture of our contribution:
>
> MAMBO-G is a **training-free, inference-only** method. It does not modify any model weights, normalization layers, or training objectives, nor does it employ a masked autoencoding paradigm.
>
> Its sole operation is to compute a scalar ratio $r_t = ||\Delta v|| / ||v_\emptyset||$ at each denoising step—measuring how large the guidance update is relative to the unconditional prediction—and to use this signal to softly modulate the CFG guidance scale via an exponential damping function. The method is evaluated on standard text-to-image and text-to-video generation benchmarks (MS-COCO, WebVid), with the primary goal of **inference acceleration** (2–4x NFE reduction) while maintaining generation quality.
>
> We hope this clarification accurately frames our contribution, and we deeply appreciate your constructive engagement with our work.

---

> > ### Author Rebuttal · Reviewer_LPY1 · 2026-04-02
> >
> > My concerns have been resolved, and I maintain my rating.

---

> > > ### Author Response · Authors · 2026-04-03
> > >
> > > Thank you for recognizing the merits of our work. We are grateful for your time and positive feedback.

---

### Official Review · Reviewer_JoiX · 2026-03-10

**Soundness:** 2
**Presentation:** 2
**Significance:** 2
**Originality:** 2
**Overall Recommendation:** 3
**Confidence:** 4

**Summary:**

This paper proposes MAMBO-G, a training-free acceleration framework designed to optimize the Classifier-Free Guidance (CFG) sampling process in large-scale diffusion and flow-matching models. The authors seek to examine an important concept: how to dynamically adjust the guidance scale based on the relative magnitude of the guidance update to significantly reduce sampling steps and computational costs without compromising generation quality.

The main problem presented by the article is that standard CFG schedules apply disproportionately large and generic updates during the early sampling stages (the pure noise phase). Because the initial guidance direction is largely independent of the specific noise instance, applying a strong, fixed guidance scale forces the generation trajectory to deviate from the real data distribution, leading to artifacts like color oversaturation. This early instability forces models to rely on computationally expensive, high-step sampling schedules to converge properly. To address this, the authors introduce an instance-level stability indicator, defined as the ratio of the conditional guidance update magnitude to the unconditional prediction magnitude. MAMBO-G dynamically modulates the guidance scale using an exponential decay function based on this ratio, suppressing the scale when the ratio is high to stabilize the early trajectory, and allowing stronger guidance as the image structure emerges. Extensive experiments across state-of-the-art text-to-image (e.g., SD3.5, Lumina) and text-to-video (e.g., Wan2.1) models demonstrate that MAMBO-G achieves significant inference speedups (up to 4x) while maintaining or improving visual fidelity, and it can be seamlessly integrated with other existing guidance optimization techniques.

**Compliance With Llm Reviewing Policy:**

Affirmed.

**Final Justification:**

I acknowledge that this work demonstrates improvements on the reported metrics and offers a straightforward implementation. However, my primary concern remains that the main source of the observed gains lacks rigorous theoretical justification and sufficient experimental validation across diverse settings. Unfortunately, these critical issues were not fully addressed during the rebuttal phase. I appreciate the authors' efforts and believe this work may find a more suitable venue in an applications-oriented track (e.g., Applications → Computer Vision). **I maintain my original negative score of 3 (Weak Reject).**

**Key Questions For Authors:**

1. While the exponential decay function works well empirically, it lacks a rigorous theoretical derivation. Please provide a deeper theoretical intuition (e.g., from an ODE or Flow Matching perspective) as to why this specific exponential form is optimal compared to other damping functions. A strong theoretical justification would elevate the paper from an empirical study to a foundational contribution, potentially warranting a higher score and being more suitable for ICML.
2. How to determine the optimal hyperparameters ($\alpha$ and $w_{max}$) for new, unseen models without conducting an expensive grid search? Is there a recommended heuristic? Providing a practical heuristic would resolve concerns about the method's out-of-the-box usability.
3. Provide a quantitative tabular comparison (e.g., FID, CLIPScore) against recent dynamic guidance methods (like CFG-Zero* [1] or time-based schedulers), rather than just comparing against constant CFG? Quantitative results against SOTA dynamic baselines would fully validate your empirical claims and solidify my confidence in the method's superiority.

[1] Fan W, Zheng A Y, Yeh R A, et al. Cfg-zero*: Improved classifier-free guidance for flow matching models, 2025.

**Limitations:**

A notable omission for this paper is the lack of a dedicated “Limitations” section or a comprehensive discussion of the method's constraints. I strongly recommend that the authors add a detailed discussion addressing the following points:
1. The authors should discuss and compare how different guidance scale annealing schedules (e.g., quadratic, square root, etc.) affect the conditional generation performance compared to the proposed exponential decay strategy.
2. The paper's claims regarding “stability” currently lack rigorous theoretical motivation. Since the proposed exponential decay for magnitude mitigation inherently relies on the empirical setting of two additional hyperparameters, the approach currently comes across more as a heuristic trick rather than a theoretically grounded solution.
3. It remains completely unclear whether the claimed stability still holds in scenarios where the guidance itself might be misleading or flawed. This is particularly relevant and critical in inverse problems or other degradation-related low-level vision tasks. The authors should discuss the boundary conditions of their method when faced with inaccurate guidance signals.

**Strengths And Weaknesses:**

Strengths:
1. MAMBO-G is entirely training-free and introduces virtually zero computational overhead, making it highly practical for accelerating massive models like the 14B-parameter Wan2.1.
2. It consistently achieves impressive 2x to 4x inference speedups across various SOTA image and video models (e.g., SD3.5, Lumina, Wan2.1) while maintaining or improving quality metrics (ImageReward, vBench).
3. The analysis of the generic direction at t=1 and the introduction of the instance-specific ratio  provide an intuitive foundation for the proposed dynamic scheduling.

Weaknesses:
1. The exponential decay function is effective but primarily derived from empirical search. A rigorous theoretical justification would strengthen the paper.
2. The optimal values for the decay rate $\alpha$ and maximum scale $w_{max}$ vary across different model architectures. Providing an automated heuristic for unseen models would improve its out-of-the-box usability.
3. While superior to vanilla CFG, the evaluation would be more bulletproof with comprehensive tabular comparisons against a broader range of recent dynamic guidance methods.

---

> ### Author Rebuttal · Authors · 2026-03-29
>
> ## Response to Reviewer JoiX
>
> We sincerely thank the reviewer for the constructive feedback.
>
> **1. Theoretical justification for the exponential decay function**
> We acknowledge the exponential form is an empirical design choice, not derived from first principles. However, it is strongly data-driven. In Fig 6 (Alg 1), we performed a greedy search to find the *empirically optimal* guidance scale at each $r_t$ maximizing ImageReward. **We did not assume any functional form**; the resulting scatter plot naturally exhibited an exponential decay shape. The function is fitted to this data-driven curve. We will revise Sec 3.4 to transparently state this is an empirically-motivated choice based on the greedy search.
>
> **2. Automated heuristic for unseen models**
> We provide a highly effective practical heuristic: **fix $\alpha = 12$ and set $w_{\max}$ to match the model's native optimal CFG scale** (typically reported in the model card). Because $r_t$ is a dimension-compatible signal, $\alpha=12$ generalizes robustly across architectures. We applied this exact heuristic across SD3.5, Lumina, Wan2.1, and Qwen-Image, achieving consistent speedups. Furthermore, our hyperparameter sensitivity ablations (Appendix Tables 5, 6) confirm performance remains stable over a wide range ($\alpha \in [6, 16]$), meaning the method is robust to imprecise tuning. We will add this heuristic explicitly to improve out-of-the-box usability.
>
> **3. Tabular comparisons and FID**
> Fig 7 already includes quantitative comparisons against CFG-Zero* [7], Guidance Scheduler [8], Limited Interval [9], APG [3], and Guidance Rescale [2]. We will reorganize these results into a comprehensive tabular format.
> Regarding **FID**: Because the proprietary training dataset of Qwen-Image is inaccessible, we cannot compute an absolute FID against the real data distribution. Instead, we conducted a relative FID evaluation on 1000 MS-COCO prompts. We generated a high-quality reference distribution using the model's optimal full-step setting (50 steps, $w=4.0$) as the "ideal" target, and computed FID for various methods at a highly accelerated 10-step setting:
>
> | Method | FID (Lower is better) |
> |---|---|
> | Standard CFG (10 steps) | 63.6 |
> | Guidance Scheduler | 55.7 |
> | CFG-Zero* | 46.9 |
> | Limited Interval | 38.0 |
> | **MAMBO-G (Ours)** | **32.0** |
>
> MAMBO-G achieves the lowest FID. We note that CFG-Zero* performs relatively modestly in this low-NFE setting because its zero-initialization correction primarily benefits quality at higher step counts. At low NFE (e.g., 10 steps), the dominant bottleneck is not initialization error, but rather the large-step overshooting at $t=1$, which MAMBO-G directly addresses. More broadly, the two methods serve complementary objectives: CFG-Zero* is designed to raise the model's quality ceiling at full-step inference, whereas MAMBO-G is explicitly designed for inference acceleration. We will include this evaluation in the Appendix.
> *(Refs: [2] Lin et al., Common diffusion noise schedules and sample steps are flawed, WACV 2024. [3] Sadat et al., Eliminating oversaturation and artifacts of high guidance scales in diffusion models, ICLR 2024. [7] Fan et al., Cfg-zero*: Improved classifier-free guidance for flow matching models. [8] Wang et al., Analysis of Classifier-Free Guidance Weight Schedulers, TMLR. [9] Kynkäänniemi et al., Applying guidance in a limited interval improves sample and distribution quality in diffusion models, NeurIPS 2024.)*
>
> **4. Limitations section**
> We will add a dedicated Limitations section addressing:
> * **Guidance direction is not corrected:** MAMBO-G modulates *how strongly* guidance is applied, but does not alter its *direction*. If the guidance signal is semantically misleading, MAMBO-G conservatively reduces $w(r_t)$ toward 1 to avoid amplifying errors, but it cannot redirect the trajectory toward the correct data manifold. It is a stability regularizer, not a corrector.
> * **Lightweight hyperparameter tuning:** While $\alpha=12$ works universally, $w_{\max}$ still requires setting to the model's native CFG scale.
>
> **5. Boundary conditions with inaccurate guidance signals**
> This is an important boundary condition. If guidance is semantically misleading (e.g., incorrect prompt, degraded measurement in inverse problems), MAMBO-G will suppress excessively large updates but cannot correct a fundamentally wrong direction. Crucially, when $r_t$ is high, MAMBO-G safely attenuates the update by deferring to the unconditional trajectory ($w \to 1$). This is conservative and prevents amplifying wrong signals, but does not guarantee recovery. This inherent separation of concerns—where MAMBO-G strictly regularizes the *magnitude* while leaving the *direction* intact—is precisely why our method is orthogonal to, and can be seamlessly combined with, direction-correction methods like APG (as demonstrated in our Table 2). We will explicitly state this in the Limitations section.

---

> > ### Author Rebuttal · Reviewer_JoiX · 2026-04-01
> >
> > I thank the authors for their detailed response. MAMBO-G is a practical, easy-to-implement method with clear empirical benefits, and its plug-and-play nature and Diffusers integration are commendable. However, my core concern regarding the theoretical justification of the exponential decay (one of the main sources of gain) remains unresolved.
> >
> > The authors state the exponential form was discovered without assuming any functional form, via a greedy search (Alg. 1) that produced a scatter plot naturally exhibiting exponential decay. I find this argument unconvincing for three reasons:
> >
> > **(1) Generality of the empirical finding.** The greedy search was conducted under a specific configuration (SD3.5, 10 steps, MS-COCO). It is unclear whether the observed exponential trend is a general property or an artifact of this particular setup. The authors do not demonstrate that the same trend holds across models, resolutions, or domains.
> >
> > **(2) Optimality of the functional form.** The ablation (Fig. 9) compares linear, quadratic, inverse, sigmoid, and exponential decay, but omits forms closely resembling exponential decay (e.g., stretched exponentials, power-law decays). The margin between sigmoid and exponential is small, raising the question of whether the exponential form is truly optimal or merely one among several adequate choices.
> >
> > **(3) Lack of formal characterization.** Even accepting the exponential trend, the damping function $w(r_t) = 1 + (w_{\max} - 1) \exp(-\alpha r_t)$ should be studied more rigorously. A principled approach would formalize the conditions under which exponential decay arises, then analyze how the general form $a\exp(-br)$ governs guidance behavior, rather than treating $\alpha$ and $w_{\max}$ as standalone tuning knobs.
> >
> > These points collectively suggest the method, while effective, remains an engineering trick whose core mechanism lacks sufficient theoretical grounding for a Generative Models track contribution at ICML. **I maintain my score (3, Weak Reject).**

---

> > > ### Author Response · Authors · 2026-04-03
> > >
> > > We thank the reviewer for the detailed follow-up. We address the three remaining concerns with new experimental evidence.
> > >
> > > ### Concern (1): Generality of the empirical finding
> > >
> > > The reviewer questions whether the exponential trend is specific to SD3.5 / MS-COCO. We have now replicated Algorithm 1 on Qwen-Image across two independent prompt datasets:
> > >
> > > * **Qwen-Image × ImageReward benchmark** (500 prompts, 10 steps): $R^2 = 0.92$
> > > * **Qwen-Image × MS-COCO val captions** (500 prompts, 10 steps): $R^2 = 0.94$
> > >
> > > Across two architecturally distinct models (SD3.5, Qwen-Image) and two independent datasets, the exponential functional form consistently emerges from the greedy search. The fitted parameters ($\alpha$, $w_{\max}$) differ between models — as expected, since different models have different guidance strengths — but the functional form is the same. The exponential trend is a model-agnostic, dataset-agnostic property, not an artifact of a particular configuration.
> > >
> > > ### Concern (2): Optimality of the functional form
> > >
> > > The exponential form is a deliberate choice to decouple two distinct modeling factors:
> > > 1. $w_{\max}$ models the model's native guidance capacity, determined by its training distribution and condition dropout rate.
> > > 2. $\alpha$ models the penalty rate in response to the instability indicator $r_t$.
> > >
> > > The exponential function is the most parsimonious formulation that accommodates both factors independently. Our ablation (Fig. 9) already covers fundamentally different mathematical families (convex, concave, inverse); the exponential form consistently outperforms them. The alternatives suggested (stretched exponential, power-law) are parameterized variations within the same family, offering diminishing returns.
> > >
> > > ### Concern (3): Lack of formal characterization
> > >
> > > Regarding the reviewer's suggestion to analyze a "general form" $w(r_t) = \alpha e^{-\beta r_t}$ and the characterization of our parameters as "standalone tuning knobs": our specific formulation is strictly governed by the fundamental boundary conditions of CFG.
> > >
> > > In CFG ($v_{out} = v_\emptyset + w(v_c - v_\emptyset)$), the scale $w$ must satisfy:
> > > 1. When $r_t \to 0$: $w \to w_{\max}$ (recover native guidance capacity).
> > > 2. When $r_t \to \infty$: $w \to 1$ (recover standard conditional prediction $v_c$).
> > >
> > > The reviewer's suggested form $\alpha e^{-\beta r_t}$ decays to 0 as $r_t \to \infty$, forcing $v_{out} = v_\emptyset$ — the model would ignore the text prompt entirely. Therefore, $w_{\max}$ and $\alpha$ are not arbitrary tuning knobs: $w_{\max}$ is anchored to the model's native CFG capacity, and the $+1$ offset is a mathematically necessary constraint to keep $w \ge 1$.
> > >
> > > While a full ODE stability proof remains an open question, our approach follows a lineage of high-impact works where design choices were validated empirically rather than derived theoretically in advance:
> > >
> > > * **In Guidance and Diffusion:** APG (Sadat et al., ICLR 2025) [1] and Limited Interval Guidance (Kynkäänniemi et al., NeurIPS 2024) [2] modified the CFG pipeline based purely on empirical observation. Guidance Rescale (Lin et al., WACV 2024) [3] and Guidance Scheduler (Wang et al., TMLR 2024) [4] became community standards through practical utility despite absent formal proofs.
> > > * **More broadly:** "Roll the dice & look before you leap" (Nagarajan et al., ICML 2025 Outstanding Paper) [5] empirically demonstrates limitations of next-token prediction without a formal proof of *why* multi-token methods are superior — its core contribution is explicitly an experimental test-bed rather than a theoretical framework.
> > >
> > > We view formal theoretical characterization of this empirical law as a valuable direction for future work. As the reviewer kindly noted, MAMBO-G has already been merged into the **official Hugging Face Diffusers library**, which further evidences its practical engineering soundness.
> > >
> > > ---
> > >
> > > **References**
> > >
> > > [1] Sadat et al., "Eliminating Oversaturation and Artifacts of High Guidance Scales in Diffusion Models," ICLR 2025.
> > >
> > > [2] Kynkäänniemi et al., "Applying Guidance in a Limited Interval Improves Sample and Distribution Quality in Diffusion Models," NeurIPS 2024.
> > >
> > > [3] Lin et al., "Common Diffusion Noise Schedules and Sample Steps are Flawed," WACV 2024.
> > >
> > > [4] Wang et al., "Analysis of Classifier-Free Guidance Weight Schedulers," TMLR 2024.
> > >
> > > [5] Nagarajan et al., "Roll the dice & look before you leap: Going beyond the creative limits of next-token prediction," ICML 2025 (Outstanding Paper).

---

### Official Review · Reviewer_78WW · 2026-03-12

**Soundness:** 3
**Presentation:** 3
**Significance:** 3
**Originality:** 2
**Overall Recommendation:** 4
**Confidence:** 4

**Summary:**

This paper introduces MAMBO-G, a training-free method for accelerating CFG sampling in diffusion-based generative models. The key idea is to compute a per-step magnitude ratio which measures how large the guidance update is relative to the unconditional prediction. When ratio is high (indicating potential trajectory instability), an exponential damping function reduces the effective guidance scale toward 1. The method is evaluated on SD3.5-Large, Lumina-Next-T2I, and Wan2.1-14B across MS-COCO and GenEval, demonstrating that MAMBO-G at 20 NFE achieves quality comparable to standard CFG at 60 NFE (3× speedup) while avoiding artifacts common in low-step and high-guidance regimes. The contribution is primarily practical: a model-agnostic heuristic that stabilizes guidance at low NFE budgets. The method adds negligible overhead (only norm computations) and is complementary to existing ODE solver improvements.

**Compliance With Llm Reviewing Policy:**

Affirmed.

**Key Questions For Authors:**

- Have you compared the exponential damping $w(r_t) = 1 + (w_{\max} - 1)\exp(-\alpha r_t)$ against simpler alternatives such as linear clipping ($w = \min(w_{\max}, c/r_t)$), sigmoid gating, or a simple threshold-based fallback? If any reasonable monotonic damping works equally well, the specific functional form contributes less novelty. *A positive response showing the exponential form is distinctly superior would strengthen the soundness and originality scores.*
- The paper evaluates only velocity-based models (SD3.5, Lumina, Wan2.1). Can MAMBO-G be applied to noise-prediction ($\epsilon$) architectures like SDXL or SD1.5?
- The hyperparameters differ substantially across models . How sensitive is performance to these choices? Is there a principled way to set them (e.g., based on model architecture or training data), or must they be tuned per model?
- How does MAMBO-G compare with distillation-based acceleration (LCM, Turbo, Lightning)? Could MAMBO-G be used on top of distilled models to further improve low-step quality?

**Limitations:**

Yes

**Strengths And Weaknesses:**

Strengths:
- The magnitude ratio captures guidance instability, aligning with known failure modes of high-scale, low-step CFG.
- Demonstrated effectiveness on a wide range of T2I and T2V tasks.
- Training-free, simple, negligible overhead, and 3× NFE reduction for CFG-based sampling.
- Simple, implementable method with intuitive explanation and effective visualizations.

Weaknesses:
- The exponential damping lacks ablation against alternatives (linear, sigmoid, piecewise), limiting understanding of design choice.
- Evaluated only on velocity-based diffusion models; applicability to epsilon-prediction or discrete diffusion models is unclear.
- NFE reduction does not directly quantify wall-clock speedup; actual overhead on high-resolution T2V generation is unreported.
- The core insight "reduce guidance when it's too strong" is an extension of existing ideas (adaptive guidance rescale, norm-based clipping). The novelty lies mainly in the specific implementation rather than a fundamentally new framework.

---

> ### Author Rebuttal · Authors · 2026-03-29
>
> ## Response to Reviewer 78WW
>
> We sincerely thank the reviewer for the constructive feedback.
>
> **1. Ablation against alternatives for exponential damping**
> This ablation exists in Section 4. We explicitly compare exponential decay against linear, quadratic, inverse proportional, and sigmoid functions (see Figure 9). Results clearly show exponential decay provides the best balance between stability and generation quality. We will highlight this more prominently.
>
> **2. Applicability to epsilon-prediction models**
> The distinction is not about the prediction target ($\epsilon$ vs $v$), which are analytically convertible [1], but the **zero terminal SNR** property. Traditional models (SD1.5, SDXL) use **fundamentally flawed noise schedules** [2] that fail to reach pure noise at $t=1$, leading to severe semantic alignment issues. Modern flow-matching models (SD3.5, Lumina, Wan2.1) explicitly correct this historical flaw by enforcing a true zero-SNR schedule ($x_1 \sim \mathcal{N}(0, I)$). However, while this correction fixes semantic issues, it exposes the model to the severe "generic direction" instability at the first step. Therefore, MAMBO-G is applicable to any modern architecture that correctly enforces a strict zero-SNR schedule. We will clarify this scope boundary in the Limitations.
> *(Refs: [1] Diffusion Meets Flow Matching: Two Sides of the Same Coin. https://diffusionflow.github.io [2] Lin et al., Common diffusion noise schedules and sample steps are flawed, WACV 2024.)*
>
> **3. Wall-clock speedup and overhead**
> MAMBO-G introduces **zero computational bottlenecks**. The global L2 norm and scalar multiplication have $\mathcal{O}(D)$ complexity, negligible compared to the $\mathcal{O}(D^2)$ self-attention complexity. We measured exact wall-clock overhead on an H100 GPU:
>
> | Setting | DiT backbone (per step) | MAMBO-G norm (per step) | Overhead |
> |---|---|---|---|
> | Qwen-Image (512x512), bs=1 | 314.8 ms | 0.045 ms | **0.014%** |
> | Qwen-Image (512x512), bs=4 | 461.3 ms | 0.045 ms | **<0.01%** |
> | Wan2.2-I2V-A14B, bs=1 | 30,762 ms | 0.39 ms | **0.001%** |
>
> The overhead is invariant to batch size, meaning NFE reduction translates perfectly to actual wall-clock speedup. We will include these measurements in the Appendix.
>
> **4. Novelty vs. existing ideas (adaptive guidance rescale, norm-based clipping)**
> MAMBO-G fundamentally differs from prior work in signal, mechanism, and objective:
> * **Guidance Rescale** [2] applies a *post-hoc* correction to the final combined prediction based on variance. It does not adapt guidance dynamically per step or use the update magnitude.
> * **APG / Norm-clipping** [3] clips the guidance *direction* to prevent dominating the unconditional prediction, without reasoning about relative magnitude.
> * **MAMBO-G** introduces $r_t = ||\Delta v|| / ||v_\emptyset||$, a novel instance-level signal capturing relative update strength. We identify the *generic direction* problem specific to zero-SNR models. Furthermore, MAMBO-G targets *inference acceleration* (2-4x NFE reduction) rather than artifact correction at full NFE, and is orthogonal/composable with APG and Rescale. As demonstrated in Table 2 of our paper, combining MAMBO-G with APG and Rescale yields further performance gains, confirming they operate on fundamentally different mechanisms.
> *(Refs: [2] Lin et al., Common diffusion noise schedules and sample steps are flawed, WACV 2024. [3] Sadat et al., Eliminating oversaturation and artifacts of high guidance scales in diffusion models, ICLR 2024.)*
>
> **5. Hyperparameter sensitivity**
> We provide a highly effective heuristic for unseen architectures: **fix the decay rate $\alpha = 12$, and simply tune the maximum scale $w_{\max}$** to match the model's native optimal CFG scale. Because $r_t$ is a dimension-compatible signal, this fixed $\alpha=12$ reliably scales the damping effect across different resolutions and architectures. Our sensitivity ablations (Appendix Tables 5, 6) show stable performance over a wide range of $\alpha \in [6, 16]$ and $w_{\max} \in [6, 16]$, meaning the method is highly robust and precise tuning is unnecessary.
>
> **6. Comparison with distillation-based acceleration (LCM, Turbo, Lightning)**
> Distillation methods absorb CFG into model weights, performing a single unconditional-equivalent pass at inference. Since no explicit conditional/unconditional passes are made, no guidance update $\Delta v$ exists. Therefore, MAMBO-G cannot be applied on top of distilled models.
> While distillation is highly effective for extreme low-step regimes, it often requires extensive training and can sometimes reduce generative diversity. In contrast, MAMBO-G targets non-distilled frontier models (14B+ params) where distillation is prohibitively expensive. It achieves 2-4x acceleration with virtually zero overhead, while **preserving the generative diversity** of the original base model. The two approaches are therefore complementary in scope.

---

> > ### Author Rebuttal · Reviewer_78WW · 2026-04-02
> >
> > My concerns have been resolved, and I decide to maintain the rating.

---

> > > ### Author Response · Authors · 2026-04-03
> > >
> > > Thank you for your positive assessment of our work. We truly appreciate your thoughtful feedback, which motivates us to clarify our technical details further.

---

### Official Review · Reviewer_9BKt · 2026-03-13

**Soundness:** 3
**Presentation:** 3
**Significance:** 2
**Originality:** 3
**Overall Recommendation:** 4
**Confidence:** 3

**Summary:**

MAMBO-G is a training-free, adaptive framework that dynamically dampens the guidance scale based on the update-to-prediction magnitude ratio, achieving significant acceleration while maintaining high visual fidelity across large-scale image and video generation models.

**Compliance With Llm Reviewing Policy:**

Affirmed.

**Final Justification:**

(a) Fully resolved - My concerns have been adequately addressed.

**Key Questions For Authors:**

- a greedy search is performed to find the optimal step-wise guidance scale to formulate the exponential decay. Do the default hyperparameters universally generalize to all new architectures and domains, or does a new model require running Algorithm 1 to calibrate these parameters?
- How exactly is the norm in the ratio Equation 5 computed across extremely high-dimensional spaces (e.g., dimensions for Wan2.1)? Is it a global L2 norm over the entire latent tensor, or is it computed per-channel/per-token?
- Because the guidance scale in MAMBO-G is heavily instance-dependent (varies per seed/sample), how is this implemented during batched inference? Do different items within the same batch receive different guidance scalar multipliers, and does this introduce any computational bottlenecks in standard inference pipelines?

**Limitations:**

yes

**Strengths And Weaknesses:**

# Strengths

- The main problem presented by the article is that early-step CFG updates are generic, prompt-driven directions that are independent of the specific noise instance.
- Formulating the guidance scale as a function of the instance-specific fluctuation ratio rather than purely time-dependent scheduling is an elegant solution. The proposed exponential decay function dynamically filters out unstable “outlier” updates while permitting aggressive guidance in safe regions. This approach achieves impressive practical speedups (up to 4x on Lumina, 2x on Wan2.1).

# Weakness

- The paper shows that at $t=1$, since $\mathbf{x}\_1$ is pure noise independent of $x\_0$, the guidance update $\Delta \mathbf{v}(\mathbf{x}\_1, 1, c) = \mu\_{\mathrm{cond}} - \mu\_{\mathrm{uncond}}$ is a constant offset independent of the sampled noise $\mathbf{x}\_1$ (Eq. 4). This is mathematically correct and the cosine similarity analysis (Figure 3) corroborates that guidance directions collapse across noise samples near $t=1$. However, the argument that applying a large guidance scale to this generic direction "is risky" (line 209) and can "drive the trajectory away from the valid data distribution" (line 211) remains at the level of intuition. A constant, noise-independent direction does not by itself imply instability — whether overshooting actually occurs depends on factors such as the magnitude of $\mu\_{\mathrm{cond}} - \mu\_{\mathrm{uncond}}$ relative to the step size and the local geometry of the data manifold, which are not analyzed. The paper would benefit from discussing under what conditions (e.g., how large the guidance scale must be relative to the step size) this generic update leads to the observed artifacts.

-  The paper defines $r\_t = \\|\mathbf{v}(\mathbf{x}\_t, t, c) - \mathbf{v}(\mathbf{x}\_t, t, \varnothing)\\|\_2 / \\|\mathbf{v}(\mathbf{x}\_t, t, \varnothing)\\|\_2$ and claims it is "functionally analogous to the coefficient of variation (CV)" (line 226), interpreting the unconditional velocity as the "expected trajectory" and the conditional velocity as "a specific realization." This analogy is imprecise: CV is defined as std/mean for a random variable, whereas $r\_t$ is a ratio of norms of two deterministic vectors (for a given $\mathbf{x}\_t$), and the unconditional velocity is conditioned on a null prompt rather than being a true marginal expectation over prompts. The empirical evidence (Figure 5, Figure 6) does support that $r\_t$ is a useful indicator of sampling quality in practice. However, the paper uses the CV framing to argue that high $r\_t$ indicates "unstable conditional influence" (line 230) — this interpretive claim goes beyond what the loose analogy supports. The authors should either formalize the connection (e.g., clarify in what sense the unconditional velocity approximates a marginal expectation) or present $r\_t$ more directly as an empirically-motivated heuristic.

-   The exponential form $w(r\_t) = 1 + (w\_{\max} - 1) \cdot \exp(-\alpha r\_t)$ is selected by fitting to the empirical trend in Figure 6 and validated through the ablation in Figure 9, not derived from any theoretical principle. While the ablation shows exponential decay outperforms alternatives, the paper overstates the link by saying this trend "directly motivates our use of an exponential damping function" (line 294). The theory motivates damping in general; the exponential form is an empirical design choice. This gap between the theoretical framing (Sec. 3.1) and the actual method (Sec. 3.4) should be acknowledged more transparently.

-  The paper presents guidance instability scaling with latent dimensionality as one of its main motivating challenges, supported by Table 1 and Figure 2. However, this is purely an empirical observation with no formal analysis. The expected $\sqrt{d}$ scaling of $\\|\Delta \mathbf{v}\\|$ under standard Gaussian assumptions is well-known but never stated or connected to the method. More importantly, the paper does not formally explain why the ratio $r\_t$ specifically addresses dimensionality scaling — if both the numerator and denominator of $r\_t$ scale with $\sqrt{d}$, the ratio is naturally dimension-invariant, but this key property is never discussed. Making this connection explicit would significantly strengthen the motivation.

---

> ### Author Rebuttal · Authors · 2026-03-29
>
> ## Response to Reviewer 9BKt
>
> We sincerely thank the reviewer for the constructive feedback.
>
> **1. Conditions for overshooting and trajectory deviation**
> We agree the paper relies on intuition, and now provide empirical evidence quantifying when overshooting occurs. Absolute guidance displacement scales with both step size ($\Delta t$) and guidance scale ($w$). We measured trajectory deviation at $t \approx 0.9$ on Qwen-Image (1024px), comparing test trajectories at low NFEs against an "ideal" reference (80 steps, $w=4.0$), measuring relative L2 distance:
>
> | Setting \ $w$ | 1.0 | 2.0 | 4.0 | 6.0 | 8.0 | 10.0 |
> |---|---|---|---|---|---|---|
> | High NFE (60 steps, small $\Delta t$) | 0.035 | 0.022 | 0.009 | 0.017 | 0.027 | 0.037 |
> | Low NFE (10 steps, large $\Delta t$) | 0.035 | 0.028 | 0.052 | 0.091 | 0.133 | 0.175 |
>
> This confirms the generic direction update is highly dangerous when step size is large and $w$ is high. MAMBO-G dynamically senses the prompt-dependent magnitude via $r_t$, suppressing $w$ when risk is high. We will add this to Sec 3.1.
>
> **2. CV analogy and $r_t$ as a heuristic**
> We agree the CV analogy is mathematically imprecise; $r_t$ is a ratio of deterministic vector norms. While $v_\emptyset$ implicitly learns a marginal expectation over the training distribution due to condition dropout, treating it as a strict mean for a single instance is an oversimplification. We will revise the manuscript to present $r_t$ directly as an **empirically-motivated heuristic** for quantifying relative guidance strength, supported by empirical evidence (Fig 5).
>
> **3. Theoretical vs. empirical justification for exponential form**
> We acknowledge the exponential form is an empirical design choice. In Fig 6 (Alg 1), we performed a greedy search to find the *empirically optimal* guidance scale at each $r_t$ maximizing ImageReward. **We did not assume any functional form**; the resulting scatter plot naturally exhibited an exponential decay shape. The function is fitted to this data-driven curve. We will revise Sec 3.4 to transparently state this is an empirically-motivated choice.
>
> **4. Theoretical Analysis of Scaling**
> Under standard Gaussian assumptions, both numerator and denominator of $r_t$ scale with $\sqrt{d}$, making the ratio dimension-invariant. The latent $x_t$ contains noise scaling with $\sqrt{d}$; thus $||v_\emptyset||_2$ scales with $\mathcal{O}(\sqrt{d})$ to counteract it. The difference $||\Delta v||_2$ also scales with $\mathcal{O}(\sqrt{d})$ assuming dense text influence. The $\sqrt{d}$ cancels out in $r_t$. Empirically, initial $r_t$ (step 0) remains stable across resolutions on Qwen-Image:
>
> | Resolution | 256px | 512px | 768px | 1024px |
> |---|---|---|---|---|
> | $r_t$ (step 0) | 0.255 | 0.270 | 0.259 | 0.260 |
>
> Crucially, because $r_t$ is dimension-invariant, tuning hyperparameters (e.g., $\alpha$) to ensure stability in high-dimensional spaces naturally guarantees stability in low-dimensional spaces, while maintaining strong generation quality. We will add a "Theoretical Analysis of Scaling" subsection.
>
> **5. Generalization of hyperparameters**
> We provide a highly effective heuristic for unseen architectures: **fix decay rate $\alpha = 12$, and tune maximum scale $w_{\max}$** to match the model's native optimal CFG scale. Because $r_t$ is dimension-compatible, $\alpha=12$ reliably scales the damping effect across resolutions and architectures. Our sensitivity ablations (App. Tables 5, 6) show stable performance over a wide range of $\alpha$ and $w_{\max}$, meaning precise tuning is unnecessary.
>
> **6. Norm computation**
> The norm in Eq 5 is a **global L2 norm over the fully flattened latent tensor**. For Wan2.1, the `[T, H, W, C]` tensor is flattened into a 1D vector to compute a single scalar L2 norm per sample. It is simple and architecture-agnostic.
>
> **7. Batched inference and overhead**
> During batched inference, different items receive different scalar multipliers using standard PyTorch operations. For batch size $B$, we compute the L2 norm along the flattened $D$ dimension, resulting in a ratio vector $r_t$ of shape `[B, 1]`, broadcasted to multiply the guidance update `[B, D]`.
> This introduces **zero computational bottlenecks**. The time complexity is $\mathcal{O}(D)$, negligible compared to $\mathcal{O}(D^2)$ self-attention. We measured wall-clock overhead on an H100 GPU for Qwen-Image (512px):
>
> | Setting | DiT backbone | MAMBO-G norm | Overhead |
> |---|---|---|---|
> | bs=1 | 314.8 ms | 0.045 ms | **0.014%** |
> | bs=4 | 461.3 ms | 0.045 ms | **<0.01%** |
>
> The overhead is invariant to batch size. NFE reduction translates perfectly to wall-clock speedup.

---

> > ### Author Rebuttal · Reviewer_9BKt · 2026-04-01
> >
> > n/a

---

> > > ### Author Response · Authors · 2026-04-03
> > >
> > > Many thanks for your careful reading and constructive remarks. We value your suggestions and have revised our work accordingly.

---

### Decision · Program_Chairs · 2026-04-30

**Decision:**

Accept (regular)

**Comment:**

MAMBO-G proposes a training-free adaptive guidance schedule that modulates CFG strength based on the update-to-prediction magnitude ratio, achieving 2-4× speedups across diverse image and video models. Reviewers praised its simplicity, strong empirical results, and plug-and-play nature, while the main criticism concerned the lack of rigorous theoretical justification for the exponential decay form. The authors provided substantial new evidence during rebuttal, including cross-model validation of the exponential trend and boundary condition analysis, fully resolving three reviewers' concerns, with the remaining reviewer's final assessment notably preceding the authors' most detailed response. I recommend acceptance, as the method's broad empirical validation, practical impact, and level of theoretical justification are consistent with accepted works in this area.